# HaloProbe: Bayesian Detection and Mitigation of Object Hallucinations in Vision-Language Models

**Reihaneh Zohrabi** [*1] **Hosein Hasani** [*2] **Akshita Gupta** [1] **Mahdieh Soleymani Baghshah** [2]
**Anna Rohrbach** [1] **Marcus Rohrbach** [1]

## Abstract

Large vision-language models can produce object hallucinations in image descriptions, highlighting the need for effective detection and mitigation strategies. Prior work commonly relies on the model's attention weights on visual tokens as a detection signal. We reveal that coarse-grained attention-based analysis is unreliable due to hidden confounders, specifically token position and object repetition in a description. This leads to Simpson's paradox: the attention trends reverse or disappear when statistics are aggregated. Based on this observation, we introduce HaloProbe, a Bayesian framework that factorizes external description statistics and internal decoding signals to estimate token-level hallucination probabilities. HaloProbe uses balanced training to encourage learning from internal evidence and combines it with a learned prior over external features to recover the true posterior. While intervention-based mitigation methods often degrade utility or fluency by modifying models' internals, we use HaloProbe as an external scoring signal for non-invasive mitigation. Our experiments show that HaloProbe-guided decoding reduces hallucinations more effectively than state-of-the-art intervention-based methods while preserving caption utility and linguistic fluency.

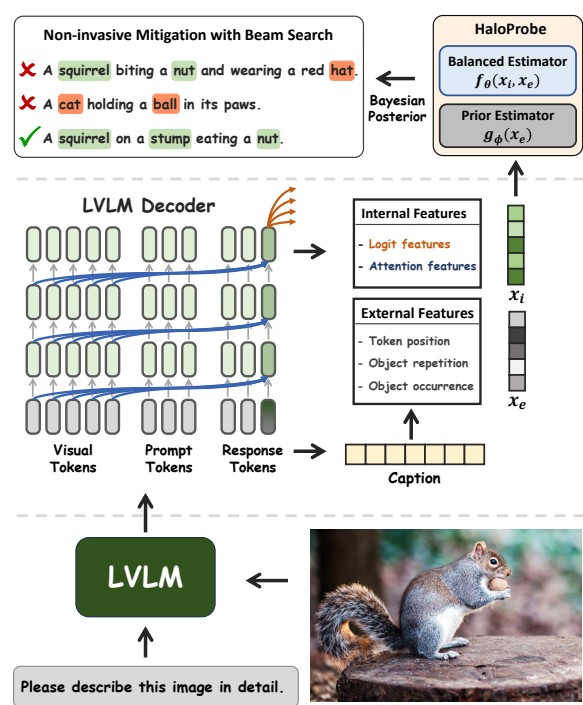

*Figure 1.* Overview of HaloProbe. Given an image and a prompt, an LVLM generates a caption. HaloProbe adopts a Bayesian formulation that combines internal features (e.g., attention and logit statistics) with external caption features (e.g., object repetition and its token position) through a balanced estimator and a prior estimator to produce token-level hallucination scores. HaloProbe enables reliable hallucination detection and downstream hallucination mitigation without modifying model internals.

## 1. Introduction

Large vision-language models (LVLMs) (Bai et al., 2025b; 2023; Liu et al., 2024a;b; Zhu et al., 2023; Chen et al., 2023) have recently gained significant attention due to their rapid

[*]Equal contribution [1]Technical University of Darmstadt & hessian.AI, Germany [2]Department of Computer Engineering, Sharif University of Technology, Tehran, Iran. Correspondence to: Reihaneh Zohrabi <reihaneh.zohrabi@tu-darmstadt.de>.

*Proceedings of the 43rd International Conference on Machine Learning*, Seoul, South Korea. PMLR 306, 2026. Copyright 2026 by the author(s).

advancements, enabling them to excel across a broad spectrum of visual perception and reasoning tasks (Shao et al., 2024; Zhou et al., 2025; Jain et al., 2025; Team et al., 2026). However, despite their strong capabilities, LVLMs often produce object hallucinations (Rohrbach et al., 2018), i.e., refer to objects not present in an image, particularly in open-ended generation tasks. This behavior reduces the overall reliability of their outputs, limiting their trustworthiness in practical applications and motivating research on hallucination detection, which identifies non-existent objects, and hallucination mitigation, which aims to prevent such errors.

Recent studies (Jiang et al., 2025; Che et al., 2025) treat image attention values of generated tokens as an indicator for distinguishing correct objects present in an image from the hallucinated ones, with (Jiang et al., 2025) specifically reporting higher image attention for correct objects. Our analysis challenges this view by revealing that two hidden confounders induce Simpson's paradox (Simpson, 1951), leading to contradictory conclusions depending on how attention statistics are aggregated. Specifically, conditioning on *generated token position* or an indicator of its *occurrence* (first versus non-first mention) produces trends conflicting with those obtained by marginalizing over these factors. In other words, *correct and hallucinated objects exhibit different token position patterns and occurrence distributions*. When these factors are ignored, attention-based analyses can overstate the reliability of global attention (averaged across layers and heads) as a predictive signal for hallucination detection.

Motivated by these observations, we introduce **HaloProbe**, a Bayesian framework for hallucination detection that incorporates token-level statistics alongside internal signals. In this framework, internal features are derived from the LVLM's dynamics, such as attention values and decoder confidence signals. External features, including token position and object repetition, capture coarse-grained statistical properties of generated captions and are therefore easy to learn. Combined with the severe class imbalance between correct and hallucinated samples, this makes models prone to shortcut learning and biased predictions. HaloProbe reduces this risk by introducing a unified probabilistic framework that factorizes learning from different types of signals, increasing robustness to unintended biases.

Recently, intervention-based hallucination mitigation methods (Qian et al., 2025; Yang et al., 2025; Jung et al., 2025; Liu et al., 2024c; Jiang et al., 2025; Che et al., 2025) have gained popularity due to their effectiveness; they rely on direct modulation of model internals, such as attention values. These approaches intervene in various ways, including targeting specific attention heads (Qian et al., 2025; Yang et al., 2025), all heads (Liu et al., 2024c) or restricted layer ranges (Jiang et al., 2025), the top-$k$ most attended image tokens based on head-averaged attention in a given layer (Che et al., 2025), or selectively and progressively recalibrating visual token attention throughout decoding (Jung et al., 2025). Further details are provided in Appendix A.

In this work, we show that intervention-based mitigation can degrade fluency and introduce unnatural generation artifacts by shifting the LVLM from its standard operating regime. This limitation underscores the importance of developing more reliable, decoding-level mitigation strategies that preserve the original generation behavior. We show that HaloProbe can be employed as an effective probe for non-invasive post-hoc mitigation methods. We design a beam search strategy that prioritizes candidate captions based on the scores estimated by HaloProbe as shown in Fig. 1; we also consider a simple post-processing mitigation scheme. Experiments on MS COCO (Lin et al., 2014) using the CHAIR metric (Rohrbach et al., 2018) show that our approach outperforms state-of-the-art methods for open-ended caption generation, while remaining non-invasive and relying solely on the standard decoding procedures of LVLMs. Importantly, this demonstrates that naturally generated responses exist within the decoded caption distribution, and an accurate probe is sufficient to identify them. This reduces the need for deploying methods that rely on interventions in LVLMs' internal dynamics, which can have unintended consequences.

Our main contributions are as follows:

- We identify token position and object occurrence as hidden confounders in attention-based hallucination analysis and show that they induce Simpson's paradox, leading to misleading conclusions when attention statistics are aggregated. We provide empirical evidence that globally averaged image attention is an unreliable signal for hallucination detection once confounding factors and class imbalance are taken into account.

- We propose **HaloProbe**, a Bayesian hallucination detection framework that disentangles internal model signals from external caption statistics using balanced training and posterior correction.

- We demonstrate that HaloProbe enables effective decoding-level hallucination mitigation via non-invasive beam search and post-hoc processing, outperforming intervention-based methods while preserving generation fluency.

## 2. HaloProbe: A Bayesian Hallucination Detection Framework

In this section, we introduce HaloProbe by describing both its motivation and technical formulation. HaloProbe is developed based on three main methodological contributions. The first key factor is conditioning on external features such as object occurrence, token position, and object repetition. This design is motivated by our analysis of Simpson's paradox, which shows that trends in marginal attention statistics disappear or reverse when conditioning on external features. Second, we use fine-grained attention signals at the level of individual layers and heads, rather than coarse-grained attention values, which are not sufficiently discriminative once conditioning is applied. Third, we introduce a Bayesian framework that naturally decomposes learning from com-

plex internal features and simpler external features. This formulation enables effective representation learning under severe class imbalance and reduces reliance on shortcut features, while retaining the predictive information contained in simple features through posterior correction rather than discarding them.

## 2.1. Problem Setup

We formulate hallucination detection as a token-level probabilistic inference problem. For a caption $c$, we treat each object token in the caption as a basic unit of analysis. Let $c_t$ denote the object token at position $t$ in caption $c$. [1]

Each token $c_t$ is associated with a correctness label $y(c_t) \in \{0, 1\}$, where $y(c_t) = 1$ indicates a correct object and $y(c_t) = 0$ indicates a hallucinated object. The token position is denoted by $t$. We use $r(c_t)$ to denote the repetition count of the corresponding object, and $o(c_t) \in \{\text{first}, \text{non-first}\}$ to indicate whether the token is the first occurrence of that object in the caption.

Each token is described by two types of features. External features $x_e(c_t)$ capture surface-level properties of the caption, including token position $t$, object repetition $r$, and occurrence $o$. Internal features $x_i(c_t)$ capture signals produced during decoding, including attention values across layers and heads and decoder confidence statistics. The goal of hallucination detection is to estimate, for each object token $c_t$, the posterior probability $p\big(y(c_t) \mid x_i(c_t), x_e(c_t)\big)$, which reflects the likelihood of an object being correct or hallucinated, given both external features and internal model signals. For notational simplicity, when the context is clear, we omit the explicit dependence on $c_t$ and write $y$, $r$, $o$, $t$, $x_i$, and $x_e$ instead of $y(c_t)$, $r(c_t)$, $o(c_t)$, $t(c_t)$, $x_i(c_t)$, and $x_e(c_t)$, respectively.

## 2.2. Dataset Bias and Hidden Confounders

A common assumption about hallucinated objects is that they are generated with lower image attention levels. This assumption is based on the intuition that hallucinated objects are mainly generated due to language model priors, without attending to image contents. However, our empirical experiments show that when considering additional factors, the coarse-grained attention analysis could be ambiguous and lead to different conclusions.

**Token position.** Prior work (Jiang et al., 2025) provided evidence that coarse-grained attention values from intermediate layers can serve as predictive signals for hallucination detection. Here, we analyze the role of token position as a

---

[1]We assume that object mentions can be identified at the token level in MS COCO captions. An object may correspond to one or multiple tokens; in the multi-token case, we retain only the first token for analysis.

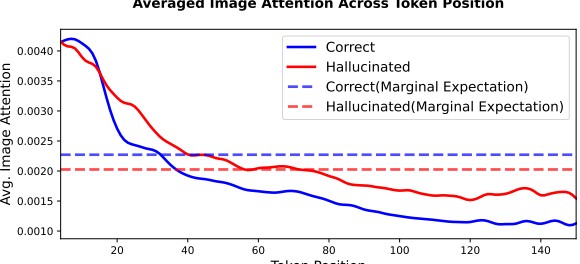

*(a)*

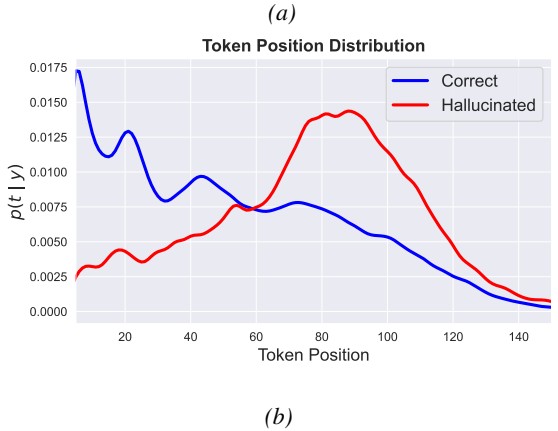

*(b)*

*Figure 2.* Illustration of Simpson's paradox induced by token position. (a) Token-position–conditioned image attention, averaged over heads, layers, and samples, for correct and hallucinated object tokens. Image attention is computed by averaging attention values from layers 5 to 18 of LLaVA-1.5-7B and over 5K samples from the MS COCO dataset. Across most positions, hallucinated tokens receive higher conditional attention than correct tokens. (b) Class-conditional token position distributions, showing that hallucinated tokens tend to appear at later positions than correct tokens. When conditioning is removed by marginalizing over token position using the distributions in (b), the expected attention values (dashed lines in (a)) reverse, with correct object tokens exhibiting higher overall attention.

confounding factor in such coarse-grained attention analysis. We denote by $A(c_t)$ the averaged image attention of token $c_t$, computed over intermediate layers, all attention heads, and the top-20 most attended image patches. We consider the conditional expected attention $\mathbb{E}_c[A \mid y, t]$, where the expectation is taken over object tokens $c_t$ in the dataset with fixed hallucination label $y$ and token position $t$.

Empirically, for both hallucinated and correct tokens, the expected image attention decreases as the token position increases as shown in Fig. 2a. This trend is consistent with observations reported in prior works (Jung et al., 2025; Liu et al., 2024c). At the same time, the position distributions differ across labels: correct objects tend to appear earlier in captions, while hallucinated objects are more likely to occur at later positions, Fig. 2b. Contrary to the common assumption that correct objects receive higher image attention, when conditioning on token position, hallucinated tokens often exhibit comparable or higher image attention than

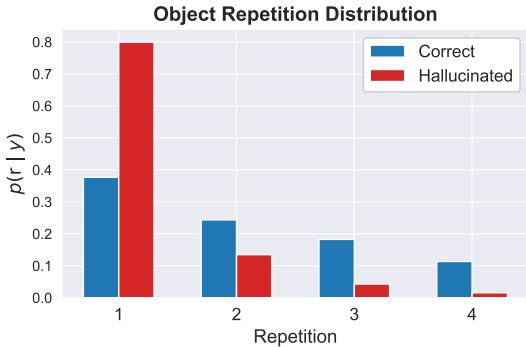

Figure 3. Distribution of object repetition counts ($r \in \{1, 2, 3, 4\}$) conditioned on class. Hallucinated objects are typically mentioned only once, while correct objects are more frequently repeated within a caption.

correct tokens (Fig. 2a):

$$\mathbb{E}[A \mid y = 0, t] \geq \mathbb{E}[A \mid y = 1, t] \quad \text{for most } t.$$

Yet, when marginalizing over token position, we obtain

$$\mathbb{E}_{c,t}[A \mid y] = \sum_t p(t \mid y)\, \mathbb{E}_c[A \mid y, t],$$

which yields the opposite trend

$$\mathbb{E}_c[A \mid y = 1] > \mathbb{E}_c[A \mid y = 0],$$

due to the different weighting induced by $p(t \mid y)$ (Fig. 2). This reversal is a clear instance of Simpson's paradox, showing that naive coarse-grained attention comparisons that ignore token position can be misleading.

**Object occurrence/repetition.** A second confounding factor arises from object occurrence/repetition. Let $o \in \{\text{first}, \text{non-first}\}$ denote whether an object mention is the first occurrence in the caption.

First-occurring tokens tend to attend more strongly to the image, as illustrated in Fig. 4a. Moreover, correct objects are repeated more frequently than hallucinated ones (Fig. 3), which reduces their probability of being first occurrences:

$$p(o = \text{first} \mid y = 0) > p(o = \text{first} \mid y = 1).$$

When attention is averaged over all object mentions (Fig. 2a), this repetition imbalance affects the marginal expected attention:

$$\mathbb{E}_{c,o}[A \mid y, t] = \sum_o p(o \mid y, t)\, \mathbb{E}_c[A \mid y, o, t].$$

However, as shown in Fig. 4a, when conditioning on object occurrence, the attention difference between correct and hallucinated objects largely disappears:

$$\mathbb{E}[A \mid y = 1, o = \text{first}, t] \approx \mathbb{E}[A \mid y = 0, o = \text{first}, t].$$

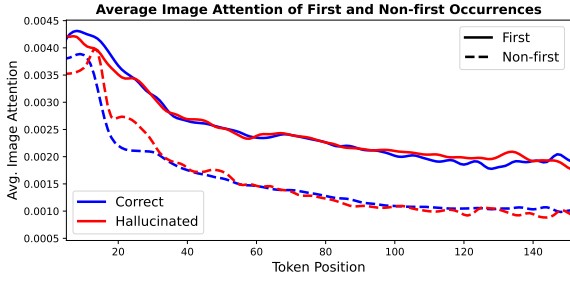

*(a)*

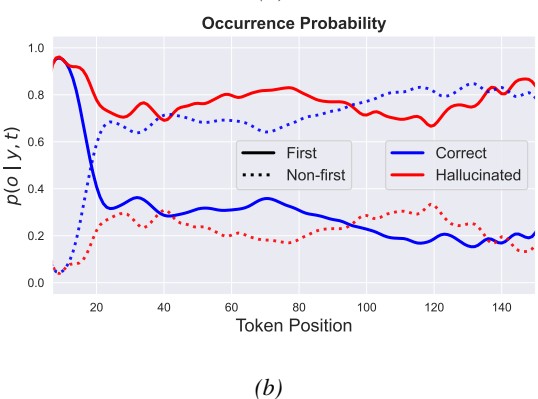

*(b)*

Figure 4. Illustration of Simpson's paradox induced by object occurrence. (a) Token-position-conditioned image attention for correct and hallucinated object tokens, shown separately for first and non-first occurrences. First mentions consistently exhibit higher image attention, even when the object is hallucinated, while non-first mentions attend less to the image. Conditioning on object occurrence largely removes the apparent attention gap between correct and hallucinated tokens. (b) Class-conditional probability of first occurrence as a function of token position, showing that hallucinated objects are more likely to appear as first mentions.

The distributions of external features (Fig. 2b for $t$, Fig. 3 for $r$, and Fig. 4b for $o$) show clear separation between hallucinated and correct classes, making them predictive features when the training and test distributions are aligned. Moreover, our Simpson's paradox analysis indicates that these features are not optional: *ignoring them may lead to incorrect conclusions*.

**Class imbalance.** Another dataset-dependent factor that strongly affects hallucination detection is class imbalance. Hallucinated objects constitute a small fraction of tokens at most positions, making them severely under-represented. Fig. 5 shows a pronounced imbalance between correct and hallucinated objects, especially at early token positions. Under this distribution, a trivial classifier that ignores the input and predicts all tokens as correct can achieve over $84\%$ accuracy with a low cross-entropy loss.

## 2.3. Balanced Training Setup

The results in the previous section show that token position and object repetition act as hidden confounders in attention-

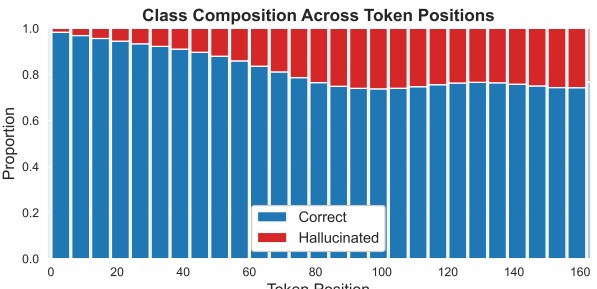

*Figure 5.* Proportion of correct versus hallucinated objects across token positions in the 5K random samples of MS COCO dataset. The dataset is highly imbalanced, particularly at early token positions.

based hallucination analysis. Ignoring these factors can lead to incorrect conclusions about the relationship between image attention and hallucination. These factors largely correspond to external features $x_e$. While omitting them degrades classifier performance, naively conditioning on these features also carries risks. Because they are easy to learn, estimators may rely on them as shortcuts, leading to biased predictions and poor utilization of internal representations. Severe class imbalance (Fig. 5) is another source of bias that can negatively impact representation learning and should be considered when designing the training strategy.

Considering these points, we train the main estimator $f_\theta$ on a dataset that is class-balanced with respect to $x_e$ (this can be done by upsampling the underrepresented class while conditioned on $x_e$). More formally, $p^{\mathrm{bal}}(y = 1 \mid x_e) = p^{\mathrm{bal}}(y = 0 \mid x_e) = \frac{1}{2}$. In this setting, the estimator learns the balanced posterior probability

$$f_\theta(x_i, x_e) := p_\theta^{\mathrm{bal}}(y = 1 \mid x_i, x_e).$$

Our main goal is to estimate the true posterior probability $p(y \mid x_i, x_e)$, while $f_\theta$ estimates a posterior under a conditional class-balanced distribution. In the following, we show how $f_\theta$ can be used to recover the true posterior.

By Bayes' rule, the posterior learned under the balanced distribution can be written as

$$p^{\mathrm{bal}}(y \mid x_i, x_e) = \frac{p(x_i \mid y, x_e)\, p^{\mathrm{bal}}(y \mid x_e)}{\sum_{j \in \{0,1\}} p(x_i \mid y = j, x_e)\, p^{\mathrm{bal}}(y = j \mid x_e)}.$$

Using the fact that $p^{\mathrm{bal}}(y \mid x_e) = \frac{1}{2}$ for both classes, we obtain

$$p^{\mathrm{bal}}(y = 1 \mid x_i, x_e) = \frac{p(x_i \mid y = 1, x_e)}{\sum_{j \in \{0,1\}} p(x_i \mid y = j, x_e)}.$$

This implies that the balanced classifier output satisfies

$$f_\theta(x_i, x_e) = \frac{p(x_i \mid y = 1, x_e)}{p(x_i \mid y = 0, x_e) + p(x_i \mid y = 1, x_e)}.$$

Rearranging terms yields an explicit likelihood ratio:

$$\frac{p(x_i \mid y = 1, x_e)}{p(x_i \mid y = 0, x_e)} = \frac{f_\theta(x_i, x_e)}{1 - f_\theta(x_i, x_e)}.$$

Thus, although $f_\theta$ is a discriminative classifier, when trained on balanced data it implicitly estimates a likelihood ratio relating internal features to the hallucination label, conditioned on external features.

### 2.4. Recovering the True Posterior

To recover the true posterior, we must account for the true label distribution conditioned on external features. We therefore train a separate model to estimate

$$g_\phi(x_e) := p_\phi(y = 1 \mid x_e),$$

using the natural, imbalanced data distribution.

This prior estimator captures how external signals alone correlate with object correctness.. In this way, we explicitly disentangle learning from easy (shortcut) features and more complex internal features. Note that $x_e$ cannot act as a shortcut during the training of $f_\theta(x_e, x_i)$, since it is no longer predictive under the balanced training setting.

We now derive the true posterior $p(y \mid x_i, x_e)$. By Bayes' rule,

$$p(y \mid x_i, x_e) = \frac{p(x_i \mid y, x_e)\, p(y \mid x_e)}{\sum_{j \in \{0,1\}} p(x_i \mid y = j, x_e)\, p(y = j \mid x_e)}.$$

Substituting the likelihood ratio derived from $f_\theta$ and the prior $g_\phi$, we obtain

$$p(y = 1 \mid x_i, x_e) = \frac{\frac{f_\theta}{1 - f_\theta}\, g_\phi}{\frac{f_\theta}{1 - f_\theta}\, g_\phi + (1 - g_\phi)},$$

where $f_\theta = f_\theta(x_i, x_e)$ and $g_\phi = g_\phi(x_e)$.

Multiplying numerator and denominator by $(1 - f_\theta)$ yields the final expression

$$p(y = 1 \mid x_i, x_e) = \frac{f_\theta\, g_\phi}{f_\theta\, g_\phi + (1 - f_\theta)(1 - g_\phi)}.$$

This posterior combines evidence from external and internal features while correcting for the bias introduced by balanced training. Practically, this formulation allows a simple interpretation: the balanced classifier $f_\theta$ and the prior estimator $g_\phi$ each output a probability distribution over the two classes. For each class, we multiply the corresponding probabilities from $f_\theta$ and $g_\phi$, and then normalize across classes.

The resulting Bayesian strategy yields a hallucination detector that is robust to the confounding effects identified in Section 2.2. We use this posterior probability as the correctness score for the downstream mitigation methods described in the next section.

# 3. Hallucination Mitigation via HaloProbe

## 3.1. Problem of Intervention-Based Strategies

Many recent methods aim to reduce object hallucinations in caption generation by directly modifying the internal dynamics of LVLMs. For example, they amplify image attention or suppress language priors during decoding. While such interventions can substantially reduce hallucinated objects, they often degrade fluency and introduce repetition or unnatural sentence structure, as shown in Fig. 6 and 7 and quantitatively analyzed in Appendix F.

These failures arise because direct manipulation of internal signals can push the model outside its standard operating regime, where internal representations deviate from those learned during training. Moreover, each attention head is responsible for one or more behavioral functions (Basile et al., 2026), and naively modifying a population of them to enforce grounding can lead to unintended consequences. These limitations highlight the risks of intervention-based mitigation and motivate non-invasive approaches that operate on model outputs without altering internal dynamics.

In this section, we focus on post-hoc hallucination mitigation strategies that preserve the original generation behavior and fluency. Improving quantitative benchmarks such as CHAIR (Rohrbach et al., 2018) should not come at the cost of linguistic quality. We show that HaloProbe provides a reliable token-level hallucination score that can be effectively used for intervention-free mitigation.

## 3.2. Hallucination-Aware Beam Search

Given a generated caption, HaloProbe assigns each object token a posterior correctness probability $p(y = 1 \mid x_e, x_i)$. These token-level probabilities, together with the resulting predictions, are used to guide a beam search strategy that prioritizes candidates with lower hallucination scores.

At decoding step $t$, we generate a set of beam candidates $\mathcal{B}_t = \{b_j^{(t)}\}_{j=1}^{N_{\text{beam}}}$. Each candidate is expanded via the LVLM's standard decoding procedure with softmax temperature $\tau > 0$, up to a maximum length $L_{\text{beam}}$. For each candidate $b_j$, HaloProbe determines the number of hallucinated and correct object mentions as $n_{\text{hal}}(b_j)$ and $n_{\text{corr}}(b_j)$, respectively. We further denote the sums of the associated hallucination and correctness confidence scores by $s_{\text{hal}}(b_j)$ and $s_{\text{corr}}(b_j)$, respectively. The overall hallucination score for a candidate is defined as

$$S(b_j) = n_{\text{hal}}(b_j) + p_{\text{hal}}(b_j) - \beta\big(n_{\text{corr}}(b_j) + p_{\text{corr}}(b_j)\big).$$

Here, $\beta$ is a hyperparameter that controls the trade-off between hallucination reduction and object class coverage. The confidence terms $p_{\text{hal}}(b_j)$ and $p_{\text{corr}}(b_j)$ provide a tie-breaking signal when candidates have identical discrete counts $n_{\text{hal}}(b_j)$ and $n_{\text{corr}}(b_j)$. Importantly, the decoding itself is unchanged: HaloProbe is used only as an external scoring mechanism. Once each candidate $b_j^{(t)}$ is assigned a hallucination score, the top-ranked candidate is retained and expanded at the next step, while all others are discarded. This procedure is repeated until an end-of-sequence token is produced.

## 3.3. Post-Process Hallucination Removal

While hallucination-aware beam search preserves language fluency, it increases computational cost linearly with the beam size $N_{\text{beam}}$. Moreover, this strategy is applicable only under stochastic decoding, i.e., when the softmax temperature satisfies $\tau > 0$. As an alternative post-hoc mitigation strategy, we mark hallucinated objects identified by HaloProbe using a specific marker. The marked captions are then passed to a single-step linguistic editing stage using an external LLM. The editor is instructed to remove only the marked hallucinated objects while keeping the remaining content unchanged. If removing a marked object results in awkward phrasing, minimal local edits are allowed to restore grammaticality and coherence. The editor is explicitly constrained to avoid introducing new objects or modifying unmarked content. This strategy is applicable to deterministic greedy decoding as well as nucleus sampling.

# 4. Experiments

## 4.1. Experimental Setup

In this paper, we evaluate object hallucination detection and mitigation on several widely adopted LVLMs, namely LLaVA-1.5 (Liu et al., 2024a), Shikra (Chen et al., 2023), and MiniGPT-4 (Zhu et al., 2023). To assess generalization to more recent architectures, we additionally include newer models such as Qwen3-VL (Bai et al., 2025a) and InternVL3.5 (Wang et al., 2025). For consistency, we use the 7B- or 8B-parameter variant of each model where applicable. Our experiments focus on open-ended caption generation and object hallucination. Additional results on discriminative settings, including attribute and relation hallucination benchmarks, are provided in Appendix J.

For all analyses and for training our hallucination detection framework, we randomly sample 5k images from the MS COCO 2014 validation set (Lin et al., 2014). The sampled set was divided equally into training and test splits for model training. To evaluate the hallucination mitigation part, we additionally sample 500 disjoint images from the same MS COCO validation set. For each image, we generate a single detailed caption using the unified prompt, *"Please describe this image in detail."*, and identify hallucinated and correct object mentions using the CHAIR (Rohrbach et al., 2018) evaluation framework (details are in Appendix B).

*Table 1.* Performance comparison of different object hallucination detection methods on LLaVA-1.5-7B backbone. Missing values are indicated with "–".

| Method | Acc. ↑ | AUROC ↑ | Precision ↑ | Recall ↑ | F1 ↑ |
|---|---|---|---|---|---|
| IC | 62.56 | – | 61.93 | 81.60 | 70.42 |
| UT | 50.57 | – | 53.60 | 70.62 | 60.95 |
| EAZY | 78.77 | – | 78.41 | 83.38 | 80.82 |
| DIML | 84.46 | 90.19 | – | 72.34 | – |
| **HaloProbe** | **90.00** | **93.50** | **92.50** | **95.80** | **94.10** |

### 4.2. Object Hallucination Detection

**Implementation Details.** We detect hallucinated object mentions at the token level using a two-layer MLP that combines internal and external features as a balanced estimator. The model outputs per-token class probabilities, which are refined using a one-layer prior network over token position and repetition count to capture structural biases. Both networks are trained with the Adam optimizer (learning rate 1e-3, weight decay 1e-3) for 10 epochs, using a batch size of 128. Further details on the input features for each network are provided in the Appendix C.

**Metrics.** Object hallucination detection is formulated as a binary classification problem, where each token is labeled as either correct (positive class) or hallucinated (negative class). To evaluate the detector, we report standard metrics including accuracy, AUROC, precision, recall, and F1 score.

**Baselines.** We compare our detection framework with recent baselines (Jiang et al., 2025; Che et al., 2025; Zhou et al., 2023a; Jiang et al., 2024b). EAZY (Che et al., 2025) reports detection results on 200 Hall-COCO images, a subset of MS COCO specifically curated to induce object hallucinations, and includes comparisons with UT (Zhou et al., 2023a) and IC (Jiang et al., 2024b). DIML[2] (Jiang et al., 2025) reports results on 500 random MS COCO samples. All methods were evaluated using the same models, and we report the published numbers from these papers for comparison. Our evaluation uses a separate set of 500 random COCO samples with the same model. Given that all sets are drawn from the same underlying distribution, our results are directly comparable to prior works and provide a statistically reliable measure of detection performance.

**Results.** Table 1 compares object hallucination detection performance across recent baselines using the LLaVA-1.5-7B backbone. Our proposed HaloProbe consistently outperforms prior approaches on all reported metrics. In particular, HaloProbe improves upon Devils-in-middle-layers (DIML) (Jiang et al., 2025), the previous state-of-the-art, by over 5 points in accuracy and over 3 points in AUROC, while also showing a significant margin of improvement

---

[2]Our abbreviation for the paper "Devils in Middle Layers of Large Vision-Language Models."

in both precision and recall. These results demonstrate the effectiveness of HaloProbe's integration of internal and external features, combined with its balanced training and prior estimation strategies, in delivering more reliable and discriminative detection of hallucinated objects. Overall, HaloProbe sets a new state-of-the-art for object hallucination detection on this benchmark.

### 4.3. Object Hallucination Mitigation

**Implementation Details.** For post-processing, we first generate the response using the LVLM. Next, we extract the objects from the response and apply HaloProbe to classify each object token as either correct or hallucinated. Hallucinated objects are then marked with a $ sign. This annotated response is provided to the GPT-5 (Singh et al., 2025) model, which is prompted to refine and edit the caption by removing only the marked objects, without making any other changes. The exact prompt used for this step is provided in the Appendix E. For our implementation of beam search, we use a beam width $N_{\text{beam}} = 5$, a temperature $\tau = 0.5$, and a $\beta = 0.1$, selecting the best beam after every $L_{\text{beam}} = 20$ tokens generated.

**Baselines.** For comparison with existing baselines, we consider three widely used decoding strategies: greedy decoding, beam search, and nucleus sampling, and evaluate our method with each. In our experiments, we use the standard implementations of these strategies as baselines. Specifically, greedy decoding selects the token with the highest probability at each step, beam search maintains multiple candidate sequences (beams) during generation and selects the sequence with the highest cumulative probability as the final output, and nucleus sampling introduces stochasticity by sampling from the top portion of the probability distribution. In addition, we compare our method against six state-of-the-art object hallucination mitigation approaches: PAI (Liu et al., 2024c), DIML (Jiang et al., 2025), EAZY (Che et al., 2025), ADHH (Yang et al., 2025), AllPath (Qian et al., 2025), and OPERA (Huang et al., 2024), which span these decoding strategies.

To ensure a fair comparison, we report results from previous works directly from their papers, as all methods were evaluated on random COCO subsets under comparable experimental settings. We verified that their reported results were consistent with our baseline; results that were not comparable were excluded. Additionally, since we evaluate three decoding strategies and PAI (Liu et al., 2024c) is applicable to all three, we reproduce PAI (marked with * in Table 2) using their official implementation under our experimental settings to ensure consistency, as some results differed across strategies.

**Benchmark and Metrics.** Following prior works (Huang et al., 2024; Che et al., 2025; Jiang et al., 2025; Liu et al.,

*Table 2.* Comparison of mitigation methods across different decoding strategies on three vision-language models. Lower $C_s$ and $C_i$ values and higher F1 scores indicate better performance. The best results are shown in **bold**, and the second-best results are underlined."–" indicates unavailable or non-comparable results and "*" indicates reproduced results.

| Decoding | Method | LLaVA-1.5 | | | Shikra | | | MiniGPT-4 | | |
|---|---|---|---|---|---|---|---|---|---|---|
| | | $C_s\downarrow$ | $C_i\downarrow$ | F1$\uparrow$ | $C_s\downarrow$ | $C_i\downarrow$ | F1$\uparrow$ | $C_s\downarrow$ | $C_i\downarrow$ | F1$\uparrow$ |
| **Nucleus** | Baseline | 53.0 | 15.2 | 74.2 | 54.4 | 16.0 | 72.3 | 30.4 | 10.4 | 68.7 |
| | PAI (Liu et al., 2024c)* | 42.0 | 13.1 | 72.0 | 50.0 | 14.6 | **73.6** | 28.6 | 9.7 | 67.2 |
| | **HaloProbe + Post-process** | **15.6** | **4.2** | **75.4** | **13.2** | **4.3** | 72.5 | **10.8** | **3.7** | **68.8** |
| **Greedy** | Baseline | 51.6 | 15.2 | 75.1 | 53.0 | 15.9 | 72.4 | 30.6 | 9.8 | 69.4 |
| | EAZY (Che et al., 2025) | 38.8 | 11.4 | - | 26.6 | 8.9 | - | - | - | - |
| | PAI (Liu et al., 2024c)* | 34.5 | 9.1 | 76.0 | 49.9 | 13.9 | **74.7** | 29.3 | 9.3 | 68.6 |
| | AD-HH (Yang et al., 2025) | 29.6 | 8.0 | - | - | - | - | - | - | - |
| | AllPath (Qian et al., 2025) | 26.6 | 7.2 | - | - | - | - | - | - | - |
| | DIML (Jiang et al., 2025) | 25.0 | 6.7 | **76.1** | 23.8 | 9.4 | 72.7 | 21.4 | 8.0 | **70.8** |
| | **HaloProbe + Post-process** | **17.6** | **5.2** | 75.2 | **15.6** | **5.0** | 73.4 | **11.8** | **4.1** | 70.3 |
| **Beam** | Baseline | 52.0 | 15.6 | 74.6 | 44.2 | 13.6 | 74.5 | 31.6 | 10.5 | 69.2 |
| | OPERA (Huang et al., 2024) | 44.6 | 12.8 | - | 36.2 | 12.1 | - | 26.2 | 9.5 | - |
| | PAI (Liu et al., 2024c)* | 33.5 | 9.4 | 75.8 | 48.0 | 13.2 | **74.9** | 31.8 | 10.5 | 69.2 |
| | **HaloProbe + Beam** | **25.2** | **7.2** | **76.1** | **19.6** | **5.8** | 74.4 | **10.3** | **4.1** | **69.7** |

(2024c), we randomly select 500 images from the MS COCO 2014 (Lin et al., 2014) validation set for the open-ended image description task.

To evaluate object hallucination in generated captions, we adopt CHAIR (Rohrbach et al., 2018) metrics, which measure hallucination at both the sentence level ($C_s$) and the instance level ($C_i$). Further details on these metrics are provided in Appendix D. In addition to CHAIR, we also report the F1 score, as it provides a more balanced view of object hallucination in LVLMs. False positives in LVLM outputs, which are largely caused by hallucinations, reduce precision, reflecting the extent of hallucinations in generated captions. Recall, on the other hand, indicates how well the model covers the set of ground-truth objects when describing an image. Since there is an inherent tradeoff between precision and recall, reporting F1 gives extra insight into the overall performance of LVLMs.

**Results on Common Evaluation Models.** As shown in Table 2, using HaloProbe for object hallucination mitigation is consistently the most effective approach across all three vision-language models (LLaVA-1.5, Shikra, and MiniGPT-4) and all decoding strategies (Nucleus, Greedy, and Beam). HaloProbe-based methods achieve the lowest $C_s$ and $C_i$ values, indicating a substantial reduction in hallucinated object tokens, while maintaining competitive or improved F1 scores compared to other methods. This highlights that our approach effectively mitigates hallucinations without sacrificing caption quality.

Compared to intervention-based beam search methods such as OPERA (Huang et al., 2024) and PAI (Liu et al., 2024c), our intervention-free strategy consistently reduces halluci-

*Table 3.* Mitigation results on Qwen3-VL. Lower $C_s$ and $C_i$ values and higher F1 scores indicate better performance.

| Decoding | Method | $C_s\downarrow$ | $C_i\downarrow$ | F1$\uparrow$ |
|---|---|---|---|---|
| **Nucleus** | Baseline | 25.2 | 8.4 | 74.4 |
| | **HaloProbe + Post-process** | **14.4** | **4.7** | **74.5** |
| **Greedy** | Baseline | 24.8 | 8.0 | **74.6** |
| | **HaloProbe + Post-process** | **12.2** | **4.3** | **74.6** |
| **Beam** | Baseline | 25.8 | 7.9 | 74.9 |
| | **HaloProbe + Beam** | **15.6** | **4.9** | **75.2** |

*Table 4.* Mitigation results on InternVL3.5. Lower $C_s$ and $C_i$ values and higher F1 scores indicate better performance.

| Decoding | Method | $C_s\downarrow$ | $C_i\downarrow$ | F1$\uparrow$ |
|---|---|---|---|---|
| **Nucleus** | Baseline | 33.8 | 9.6 | 73.8 |
| | **HaloProbe + Post-process** | **22.2** | **6.7** | **75.1** |
| **Greedy** | Baseline | 31.6 | 8.7 | 74.5 |
| | **HaloProbe + Post-process** | **16.6** | **5.1** | **75.7** |
| **Beam** | Baseline | 34.6 | 9.5 | 73.7 |
| | **HaloProbe + Beam** | **15.0** | **5.2** | **74.0** |

nated objects at the same beam width. Notably, these results demonstrate that *modifying internal model dynamics is not necessary for effective hallucination reduction*. Standard LVLM decoding, when guided by an external scoring signal such as HaloProbe, is sufficient to generate fluent and accurate captions. This robustness is evident across different models and decoding strategies, confirming the general applicability of our method. We illustrate some qualitative results in the Appendix L.

*Table 5.* Ablation of HaloProbe components on the hallucination mitigation task using the LLaVA-1.5 model. We remove fine-grained attention features, external caption features, and the Bayesian decomposition, and evaluate their impact under three mitigation settings: HaloProbe-guided beam search and post-processing (PP) applied to captions generated with greedy and nucleus decoding. Performance is reported using CHAIR metrics, where lower values indicate fewer hallucinated objects.

| Fine-grained Attention | External Features | Bayesian Decomposition | Beam | | PP – Greedy | | PP – Nucleus | | Average | |
|:---:|:---:|:---:|:---:|:---:|:---:|:---:|:---:|:---:|:---:|:---:|
| | | | $C_s\downarrow$ | $C_i\downarrow$ | $C_s\downarrow$ | $C_i\downarrow$ | $C_s\downarrow$ | $C_i\downarrow$ | $C_s\downarrow$ | $C_i\downarrow$ |
| $\times$ | $\checkmark$ | $\checkmark$ | 26.4 | 7.3 | 20.0 | 6.8 | 19.8 | 6.9 | 22.0 | 7.0 |
| $\checkmark$ | $\times$ | $\checkmark$ | 29.2 | 8.3 | 37.4 | 10.2 | 41.8 | 12 | 36.1 | 10.1 |
| $\checkmark$ | $\checkmark$ | $\times$ | 25.5 | 7.4 | 28.8 | 7.7 | 23.2 | 6.2 | 25.8 | 7.1 |
| $\checkmark$ | $\checkmark$ | $\checkmark$ | **25.2** | **7.2** | **17.6** | **5.2** | **15.6** | **4.2** | **19.4** | **5.5** |

**Results on More Recent LVLMs.** One limitation of prior evaluation studies is the reliance on relatively older LVLMs, which tend to exhibit higher hallucination rates. To assess the generality of HaloProbe, we additionally evaluate it on more recent and stronger models, including Qwen3-VL(Bai et al., 2025a) and InternVL3.5(Wang et al., 2025). Tables 3 and 4 show that HaloProbe consistently reduces hallucination across all decoding strategies, even when applied to these stronger baselines. In particular, we observe substantial improvements in both $C_s$ and $C_i$ metrics, while maintaining or slightly improving F1 scores.

**Ablation Analysis of HaloProbe.** We evaluate the contribution of the three main components of HaloProbe on the downstream hallucination mitigation task. Table 5 reports CHAIR scores when ablating fine-grained attention features, external caption features, and the Bayesian decomposition. Each configuration is evaluated under HaloProbe-guided beam search and post-processing (PP) with greedy and nucleus decoding. The full model consistently achieves the lowest hallucination rates across all settings. Removing external features leads to the largest degradation, particularly for post-processing, confirming that token position and repetition provide complementary predictive signals for hallucination detection. Removing fine-grained attention features also degrades performance, indicating that internal decoding signals remain informative when used at the head and layer level. Finally, removing the Bayesian decomposition (i.e., training $f_\theta$ on the imbalanced dataset and discarding the prior $g_\phi$) while keeping both feature groups also degrades performance, showing that factorized learning improves the reliability of hallucination scores. Overall, the results show that all three components contribute to effective mitigation. Additional ablation studies focused on the hallucination detection setting are provided in Appendix I.

## 5. Conclusion

In this work, we studied the object hallucination problem in LVLMs and showed that coarse-grained attention-based analyses are prone to hidden confounders. Token position and object occurrence induce Simpson's paradoxes, leading to misleading conclusions when attention statistics are aggregated. While these observations motivate taking external features into account, naively conditioning a detection estimator on these simple features increases the risk of shortcut learning. To address this issue, we proposed HaloProbe, a factorized Bayesian framework that disentangles internal model signals from external caption statistics. HaloProbe leverages fine-grained internal signals, including layer- and head-level attention patterns and decoder confidence statistics, rather than relying on coarse aggregated attention values. By combining conditional class balancing with posterior correction, HaloProbe produces reliable token-level hallucination scores.

HaloProbe enables effective, non-invasive hallucination mitigation through decoding-level beam search and post-hoc editing. Across multiple LVLMs and decoding strategies, HaloProbe-based mitigation strategies consistently reduce hallucinated objects while maintaining overall caption quality. These non-invasive mitigation results also suggest that captions with fewer hallucinations can be generated within the model's natural decoding behavior, requiring only an accurate probe to identify and reduce hallucinated generations. Beyond hallucination detection and mitigation, the proposed factorized Bayesian framework provides a useful perspective for studying spurious correlations, dataset biases, and fairness-related issues beyond hallucination detection.

## Impact Statement

This work studies object hallucination in LVLMs and proposes a probabilistic framework for hallucination detection and mitigation. By improving the reliability and interpretability of model outputs without intervening in internal model dynamics, the proposed approach aims to support safer and more trustworthy deployment of vision-language systems in real-world applications. The techniques introduced in this paper are intended to reduce incorrect visual descriptions and do not introduce new capabilities that raise immediate ethical concerns beyond those commonly associated with large-scale machine learning models. We do not foresee significant negative societal impacts arising specifically from this work.

## Acknowledgments

The research at TU Darmstadt was partially funded by an Alexander von Humboldt Professorship in Multimodal Reliable AI, sponsored by Germany's Federal Ministry for Education and Research. For compute, R.Z. gratefully acknowledges support from the hessian.AI Service Center (funded by the Federal Ministry of Research, Technology and Space, BMFTR, grant no. 16IS22091) and the hessian.AI Innovation Lab (funded by the Hessian Ministry for Digital Strategy and Innovation, grant no. S-DIW04/0013/003).

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

# A. Related Work

Modern large vision-language models (LVLMs), such as MiniGPT-4 (Zhu et al., 2023), LLaVA-1.5 (Liu et al., 2024a), and Shikra (Chen et al., 2023), combine powerful vision encoders (CLIP (Radford et al., 2021), EVA (Fang et al., 2023)) with pretrained language models (LLaMA (Touvron et al., 2023a;b), Vicuna (Chiang et al., 2023)) to tackle complex vision-language tasks. Despite their strong capabilities, they remain prone to object hallucinations (Zhou et al., 2023b), particularly in open-ended generation tasks (Kaul et al., 2025). In this section, we review recent attempts to detect and mitigate object hallucination in LVLMs.

**Object Hallucination Detection.** In object hallucination detection, the goal is to identify object tokens mentioned by the model that are not present in the image. Recent work, such as Internal Confidence (IC) (Jiang et al., 2024b), applies a logit lens to image hidden states and flags hallucinated objects whose maximum internal confidence is high despite being absent from ground-truth annotations. Another detection method is uncertainty (UT), which detects hallucinated tokens based on the finding that objects with higher uncertainty scores during generation are more likely to be hallucinated (Zhou et al., 2023a). Additionally, EAZY (Che et al., 2025) detects hallucinations by extracting object tokens from the generated text, tracing the top-$k$ attended image tokens for each object, and zeroing them out; if the object then disappears, it is classified as hallucinated. Moreover, Jiang et al. (2025) train a two-layer MLP on concatenated image-token attention sums across all heads and a specified layer range, using this representation to detect hallucinated objects.

**Object Hallucination Mitigation.** Existing mitigation strategies can be broadly grouped into training-based (Jiang et al., 2024a) and training-free approaches. In this work, we focus on training-free methods, which can be further categorized into: (i) attention-based methods that intervene on either the visual or textual attention mechanisms (Qian et al., 2025; Yang et al., 2025; Jung et al., 2025; Liu et al., 2024c; Jiang et al., 2025; Che et al., 2025), (ii) decoding-level controls that adjust token selection during generation (Leng et al., 2024; Huang et al., 2024; Petryk et al., 2024), and (iii) post-hoc refinement methods that rescore or enhance outputs after generation (Zhou et al., 2023a; Che et al., 2025).

In attention-based intervention approaches, methods such as AllPath (Qian et al., 2025) and ADHH (Yang et al., 2025) explicitly detect and manipulate specific attention heads that are found to promote hallucinations, either by zeroing out or scaling their values. Jiang et al. (2025) shift attention of all heads in selected mid-to-late layers by enhancing regions consistently highlighted across heads. PAI (Liu et al., 2024c) intervenes on all heads while letting the model's original attention strengths determine the modifications. EAZY (Che et al., 2025) detects hallucinatory image tokens and zeroes them out during inference to reduce hallucinations.

In decoding-based strategies, methods modify token selection during generation. OPERA (Huang et al., 2024) introduces penalties and re-ranking mechanisms within beam search to prevent the model from over-trusting ungrounded predictions. Meanwhile, VCD (Leng et al., 2024) adopts a contrastive decoding approach, comparing outputs derived from original versus perturbed visual inputs and favoring those better aligned with the true image content.

Finally, post-hoc refinement methods aim to identify and correct hallucinations after generation. LURE (Zhou et al., 2023a) performs post-generation analysis based on object co-occurrence, positional cues, and uncertainty, then rewrites the output to reduce hallucinatory content. Overall, these methods illustrate a range of strategies for mitigating hallucinations in LVLMs, from internal attention manipulation to decoding-level modifications, each with trade-offs in effectiveness and fluency.

# B. Detailed Experimental Setup and Token-Level Feature Extraction and Alignment

Across all experiments, we set the maximum generation length to 512 tokens. For each generated caption, we first apply the CHAIR evaluator to identify correct and hallucinated object mentions by aligning the caption to MS COCO instance annotations. CHAIR labels object words as correct if present in the image and as hallucinated otherwise. Using these signals, we extract internal (image attentions, scores) and external token-level features (position, and repetition), for both correct and hallucinated object tokens. These features are then used as input to our hallucination detection framework. CHAIR provides word-level labels, which we subsequently align to the model's subword tokens in order to extract token-level features.

**Token-Word Alignment.** Captions are tokenized and lemmatized to obtain a normalized word representation along with character-level offsets. Each word labeled by CHAIR is mapped to the index of its first corresponding token in the generated sequence. For earlier vision-language models such as LLaVA, Shikra, and MiniGPT-4, the generated outputs are composed of subword tokens, so a single word may correspond to multiple tokens; in these cases, the word is aligned to the index of its first constituent subword token. In contrast, more recent models such as Qwen3-VL and InternVL 3.5 exhibit a near

one-to-one correspondence between words and generated tokens, enabling more direct word–token alignment. For words that appear multiple times, we track their repetition count and explicitly mark first occurrences, enabling analysis of repeated object mentions.

**Extracted Features.** For each aligned token, we extract a set of internal and external features that capture both the model's visual grounding behavior and its generation dynamics.

- **Internal model features.**
    - **Top-$k$ attended image patches:** Indices and attention values of the most attended image tokens, providing a compact representation of visual focus. In our experiments, we used the top-20 attended image tokens ($k = 20$).
    - **Temporal attention dynamics:** Mean and entropy of the top-k attended image tokens across all layers and heads at the current decoding step and at the next decoding step. This captures the visual focus of the model immediately before and after generating a token.
    - **Logit-based confidence signals:** Token prediction scores at the current decoding step, as well as at the following step, capturing local confidence variations. In our experiments, we use the top-100 logits to capture token-level confidence.

- **External token metadata.**
    - Token ID (first subtoken)
    - Position in the generated sequence
    - Repetition count and first-occurrence indicator

## C. Input Feature Design for Balanced and Prior Estimators

In this section, we provide a detailed overview of the input features used by both the balanced estimator and the prior estimator networks, including external token-level metadata, top-$k$ visual attention statistics, and logit-based confidence measures, along with their corresponding dimensionality using the LLaVA-1.5-7B model, as summarized in Table 6.

*Table 6.* Input features for the hallucination detection balanced estimator and the prior network. The balanced estimator uses normalized features including attention statistics, logit-based features, and token metadata. The prior network uses only repetition and token position normalized to $[0, 1]$. The reported dimensionality corresponds to the LLaVA-1.5 model, which consists of 32 layers and 32 attention heads.

| Balanced Estimator Network | Dimensionality |
| --- | --- |
| First occurrence (binary) | 1 |
| Repetition count (clipped $[1, 4]$) | 1 |
| Mean of top-20 image attentions (current decoding step, all layers $\times$ heads) | $32 \times 32$ |
| Mean of top-20 image attentions (next decoding step, all layers $\times$ heads) | $32 \times 32$ |
| Top-20 image attentions entropy (current decoding step, all layers $\times$ heads) | $32 \times 32$ |
| Top-20 image attentions entropy (next decoding step, all layers $\times$ heads) | $32 \times 32$ |
| Top-100 logit-based features: entropy, max logit, max softmax | 3 |
| Normalized token position | 1 |
| *Total input dimension (Balanced estimator network)* | **4102** |
| **Prior Estimator Network:** | |
| Repetition count | 1 |
| Normalized token position | 1 |
| *Total input dimension (Prior estimator network)* | **2** |

## D. CHAIR Metrics

The **Caption Hallucination Assessment with Image Relevance (CHAIR)** (Rohrbach et al., 2018) metric is widely used in image captioning to measure the presence of hallucinated objects in generated captions. For every image, a corresponding set of ground-truth object labels is defined, and any object mentioned in a caption that does not appear in this set is considered a hallucination.

CHAIR evaluates hallucinations along two complementary levels:

- **Instance-level** ($C_i$)**:** Measures the proportion of hallucinated objects relative to all objects mentioned in captions.

- **Sentence-level** ($C_s$)**:** Measures the proportion of captions that contain at least one hallucinated object.

Formally, they are computed as:

$$C_i = \frac{|\{\text{hallucinated objects}\}|}{|\{\text{all mentioned objects}\}|},$$

$$C_s = \frac{|\{\text{captions with hallucinated objects}\}|}{|\{\text{all captions}\}|}.$$

## E. Post-Processing Prompt Design

The prompt used to guide GPT-5 (Singh et al., 2025) in refining captions is provided below. Specifically, the prompt instructs the model to remove only the objects explicitly marked as hallucinated (denoted by a $ symbol) in the input caption, while preserving all other content, structure, and wording as much as possible. This ensures that the refinement process is strictly constrained, preventing unintended alterations to correct objects or the overall semantic meaning of the caption. The prompt is carefully designed to enforce minimal edits, focusing solely on the deletion of hallucinated elements identified by HaloProbe.

---

**Prompt Example**

```
SYSTEM_PROMPT = """You are a text-editing assistant that improves image
captions by removing hallucinated objects marked with `$` while keeping the
caption fluent and faithful."""

EDITING_PROMPT = """
**Problem Description:**

We are working on a system that generates captions for images. Sometimes, the
system may hallucinate or include objects that are not actually present in
the image. These hallucinated objects are detected and marked as false
positives (FP) using a special token `$` before the object in the caption.
For example, a hallucinated object like "$refrigerator" would appear as
`$refrigerator`.

**Your Task:**

You are given a caption that includes hallucinated objects marked with `$`
(e.g., `$refrigerator`). Your task is to remove **only** the hallucinated
objects and keep the rest of the caption intact, maintaining fluency,
context, and clarity.

**Strict Instructions:**

1. **Remove Only Hallucinated Objects:**
    - The objects marked with `$` are hallucinated, and you need to **remove
    only those hallucinated objects** from the caption. For example:
      - "The image shows a spacious studio apartment kitchen with wooden
      cabinets and $refrigerator." → "The image shows a spacious studio
      apartment kitchen with wooden cabinets."
```

```
            - Do **not** remove any objects in the sentence that are not marked
            with `$`. These should be kept as they are, since they describe actual
            objects in the image.

2. **Minimal Changes:**
            - If removing a hallucinated object causes awkward phrasing, make minimal
            edits to improve the fluency of the sentence. For example:
            - **Do not delete** entire sentence structures unless absolutely
            necessary to maintain clarity.

3. **Faithfulness to the Original Caption:**
            - Ensure that the edited caption remains **faithful** to the original
            context. Do not introduce new details, objects, or replace hallucinated
            objects with new ones (e.g., don't replace `$refrigerator` with another
            new object `microwave`).
            - The resulting text should **not lose any original meaning** or
            introduce new aspects of the scene not present in the image.

4. **Clarity and Brevity:**
            - The edited caption should be clear and concise without being overly
            terse. Do not over-edit the original content. Make sure that the edited
            text does not contain objects that are marked with $ in the input text.

5. **Output Format:**
            - Provide only the final, edited caption inside **double quotes** (`""`),
            without any additional text or explanations.

The input caption is:

"""
```

## F. Effect of Attention Intervention on Decoding Stability

While attention intervention has been proposed as a mechanism to improve grounding and reduce hallucination, directly manipulating attention weights may distort the internal token dependency structure of LVLMs. In this section, we analyze the effect of attention intervention on decoding stability, repetition, and diversity under greedy decoding.

**Experimental Setup.** We compare two generation conditions using LLaVA-1.5-7B on 500 random captions of COCO: (i) greedy decoding without attention intervention, and (ii) greedy decoding with attention intervention enabled. All prompts, images, and decoding parameters are kept identical.

**Metrics.** To quantify decoding degeneration and redundancy, we report the following metrics:

- **Caption Length** ($L$): Average number of generated tokens per caption.

- **Vocabulary Size** ($|\mathcal{V}|$): Number of unique tokens used in a caption, averaged across samples.

- **RE-$n$ (Redundancy Error)**: Measures the proportion of redundant $n$-grams:

$$\text{RE-}n = \frac{\sum_{g \in \mathcal{G}_n} \max(0, c(g) - 1)}{\sum_{g \in \mathcal{G}_n} c(g)}$$

where $\mathcal{G}_n$ denotes the set of $n$-grams and $c(g)$ their counts.

- **Rep-$n$ (Repeated $n$-gram Ratio)**: Fraction of $n$-grams that appear more than once:

$$\text{Rep-}n = \frac{\sum_{g \in \mathcal{G}_n} \mathbb{I}[c(g) > 1] \cdot c(g)}{\sum_{g \in \mathcal{G}_n} c(g)}$$

- **Distinct-$n$**: Lexical diversity defined as:

$$\text{Distinct-}n = \frac{|\mathcal{G}_n|}{\sum_{g \in \mathcal{G}_n} c(g)}$$

- **Longest Repeated Span**: Length of the longest contiguous sequence of tokens that appears more than once within a caption, capturing severe loop-style degeneration.

All metrics are computed per caption and then averaged across the dataset.

*Table 7.* Decoding stability and redundancy metrics for greedy decoding with and without attention intervention. Lower is better for redundancy metrics and repeated span length.

| Condition | Len | Vocab | RE-2 | Rep-2 | Dist-2 | Span |
|---|---|---|---|---|---|---|
| No Intervention | 91.48 | 53.76 | 0.094 | 0.165 | 0.906 | 3.23 |
| Attention Intervention | 96.18 | 49.48 | 0.154 | 0.260 | 0.846 | 6.98 |

**Results.** Attention intervention significantly degrades decoding stability despite greedy decoding. Compared to the baseline, intervention increases bigram redundancy (RE-2) by 64% and repeated bigram ratio (Rep-2) by 57%, indicating disrupted local token transitions. Phrase-level diversity decreases substantially, with a 6.6% drop in Distinct-2 and an 8% reduction in vocabulary size. Most notably, the longest repeated span more than doubles, revealing severe loop-style degeneration in the generated captions. We illustrate two case studies in Figs. 6 and 7 to highlight differences in model behavior.

Interestingly, attention intervention also increases caption length, suggesting difficulty in confidently terminating generation. Taken together, these results indicate that direct manipulation of attention weights introduces instability into the decoding process, trading off factual control for reduced fluency and expressiveness.

The observed degeneration occurs under greedy decoding, which typically suppresses repetition. This suggests that the failure mode arises from internal representation distortion rather than sampling stochasticity. Our findings highlight an important limitation of attention-based intervention methods and motivate more structured approaches that preserve decoding dynamics while improving grounding.

## G. Analysis of Image Attention Across Transformer Layers

Fig. 8 presents the averaged image attention for first-occurrence object tokens, split between early (first 10) and late (last 10) transformer layers. We observe that attention in early layers decays rapidly as generation progresses, indicating that these layers are less able to sustain focus on object tokens over time. In contrast, late layers maintain relatively stable attention across token positions. Interestingly, while early-layer attention is largely non-discriminative between correct and hallucinated tokens, late-layer attention sometimes assigns higher weights to hallucinated tokens than to correct ones. These results suggest that object hallucinations are not simply a result of insufficient attention, but may be influenced by higher-layer interactions within the transformer.

## H. HaloProbe Detection Performance

In this section, we report the detection performance of the HaloProbe across three LVLM backbones, evaluating its effectiveness in identifying hallucinated objects under different model architectures.

As shown in Table 8, our proposed HaloProbe framework demonstrates consistently strong performance across all evaluated vision-language models. It achieves high precision, recall, and F1 scores, indicating reliable detection of hallucinations

👤 **User:** Please describe this image in detail.

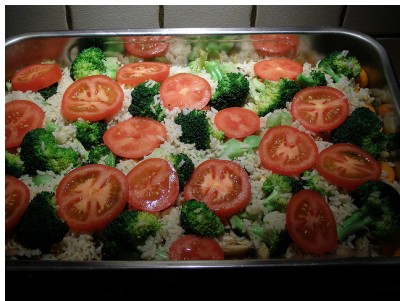

**Input Image**

🤖 **LVLM (Without Intervention)**     **RE-4 = 0.0222**

The image features a large metal pan filled with a delicious and colorful dish. The dish consists of a mixture of rice, broccoli, and tomatoes, creating a visually appealing and appetizing meal. The pan is filled with numerous pieces of broccoli, some of which are placed closer to the center of the pan, while others are scattered around the edges.

🤖 **LVLM (With Intervention)**     **RE-4 = 0.9311**

The image features a large metal pan filled with a delicious and colorful dish. The dish consists of a mix of rice, rice pilaf, and rice pilaf with rice and rice pilaf with rice and rice pilaf with rice and rice pilaf with rice and rice pilaf with rice and rice pilaf with rice and rice pilaf with rice and rice pilaf with rice and rice pilaf ...

*Figure 6.* Qualitative comparison of image description results. Given the same user prompt, the baseline model LLaVA-1.5 produces a coherent description with a low repetition score, while the intervention induces severe repetitive generation, reflected by a high RE-4 score.

*Table 8.* Performance comparison of HaloProbe across different vision-language models.

| Model | Accuracy ↑ | AUROC ↑ | Precision ↑ | Recall ↑ | F1 ↑ |
|---|---|---|---|---|---|
| LLaVA-1.5 | 90.0 | 93.5 | 92.5 | 95.8 | 94.1 |
| Shikra | 90.2 | 91.8 | 92.9 | 96.0 | 94.4 |
| MiniGPT-4 | 91.0 | 93.6 | 93.7 | 96.1 | 95.0 |

regardless of model architecture or setting. Furthermore, Figure 9 shows that HaloProbe maintains stable performance across different token positions, highlighting its robustness throughout the generated sequence. In addition, the ROC and Precision-Recall curves in Figure 10 demonstrate strong discriminative capability and reliable performance even under class imbalance, further validating the effectiveness of the proposed approach.

## I. Additional Ablation Studies for Hallucination Detection

We study the contribution of different feature groups and model components by ablating them from HaloProbe. We consider internal features derived from model dynamics, including attention weights and decoder logit-based confidence signals, as well as external features capturing token position, object repetition, and object occurrence. Excluded features are replaced with Gaussian noise. In addition to removing individual features, we also ablate entire feature groups to isolate the effect of internal and external features.

Table 9 summarizes the feature ablation results. While coarse-grained averaged attention statistics are weak predictors due to confounding, fine-grained attention patterns retain strong discriminative power. This demonstrates that image attention itself is not uninformative; rather, improper layer- and head-wise aggregation obscures its predictive signal. Ablating external features is not equivalent to ignoring the prior: since these features are replaced with Gaussian noise, the estimator $g_\phi$ effectively estimates $p(y)$, which alone improves the final posterior performance. Finally, even without any meaningful feature estimators, the model still achieves an accuracy of 84.6%, reflecting the highly imbalanced setting. Therefore, in this regime, AUROC is a more reliable evaluation metric.

👤 **User:** Please describe this image in detail.

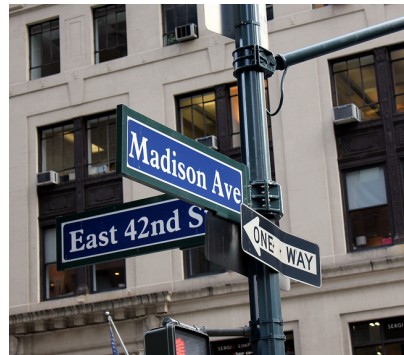

**Input Image**

🤖 **LVLM (Without Intervention)**    **RE-4 = 0.0115**

The image features a street corner with a traffic light and two street signs. The street signs are positioned above the traffic light, providing directions for drivers and pedestrians. The traffic light is located on the left side of the scene, while the street signs are on the right side.

🤖 **LVLM (With Intervention)**    **RE-4 = 0.9101**

The image features a street corner with two street signs, one for Madison Avenue and the other for East 42nd Street. The street signs are positioned on a pole, and the street signs are placed on top of a pole. The street signs are placed on a pole, and the street signs are placed on top of a pole. The street signs are placed on a pole, and the street signs are placed on top of a pole...

*Figure 7.* Qualitative comparison of image description results. Given the same user prompt, the baseline model LLaVA-1.5 produces a coherent description with a low repetition score, while the intervention induces severe repetitive generation, reflected by a high RE-4 score.

We next analyze the role of conditional class balancing in the internal feature estimator $f_\theta$. Table 10 reports the accuracy and AUROC of this estimator under different configurations of balanced training and test datasets. Compared with the full version of HaloProbe, the imbalanced classifier shows acceptable performance when the training and test distributions are identical. This highlights the effective contribution of two components of HaloProbe: internal fine-grained features and conditioning on external features. In the absence of distribution shift, the advantage of the Bayesian component is limited to providing better representation learning. To evaluate the role of this component in robust learning, we test the reliance on potential dataset shortcuts by synthetically constructing (sub-sampling) underrepresented groups from correct and hallucinated samples of the test set.

Specifically, we sample hallucinated tokens from positions 10–30 and correct object tokens from positions 110–130, corresponding to the tails of the class-conditional distributions shown in Fig. 2b. For the Bayesian estimator, the average accuracy on these minority groups is 62.3%. In contrast, for a baseline trained directly with cross-entropy loss under the true training distribution, the average accuracy on the same groups is 52.7%, which is close to that of a random binary classifier. These results indicate that factorized learning reduces reliance on shortcuts and improves robustness under distribution shifts.

## J. Additional Results on Extended Benchmarks

The main focus of this work is object hallucination, which is the standard setting in prior literature. However, the HaloProbe framework is not restricted to object-level errors. It can be extended to more complex hallucination types such as attribute and relation errors by redefining the prediction unit and feature design. In these settings, the task is no longer token-level classification, but prediction over higher-level semantic units (e.g., object-attribute pairs or object-object relations). The same factorization into internal and external features still applies.

To evaluate this, we conduct experiments on the attribute and relation subsets of the AMBER(Wang et al., 2023) benchmark using LLaVA-1.5(Liu et al., 2024a). Since AMBER is formulated as a discriminative task, we adapt HaloProbe accordingly. The model's generated response is treated as a binary external feature, and mitigation reduces to selecting the final answer based on the detector prediction. Table 11 shows that HaloProbe significantly improves performance over the baseline across all metrics, especially in recall and F1 score.

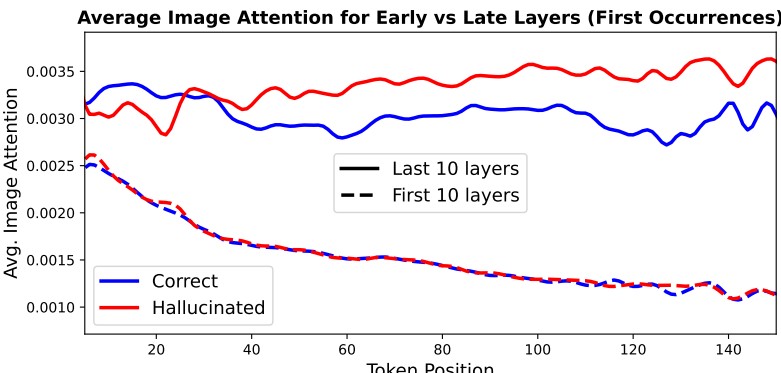

*Figure 8.* Averaged image attention for first-occurrence object tokens, averaged over early and late transformer layers. Early (first 10) layers exhibit a rapid decay in image attention as generation progresses, while late (last 10) layers maintain relatively stable attention across token positions. Attention in early layers is largely non-discriminative between correct and hallucinated tokens, whereas in late layers, hallucinated tokens counterintuitively receive higher image attention than correct tokens. Compared with Fig. 4a, this result indicates that LVLM layers behave differently and that averaging attention across all layers can obscure meaningful patterns.

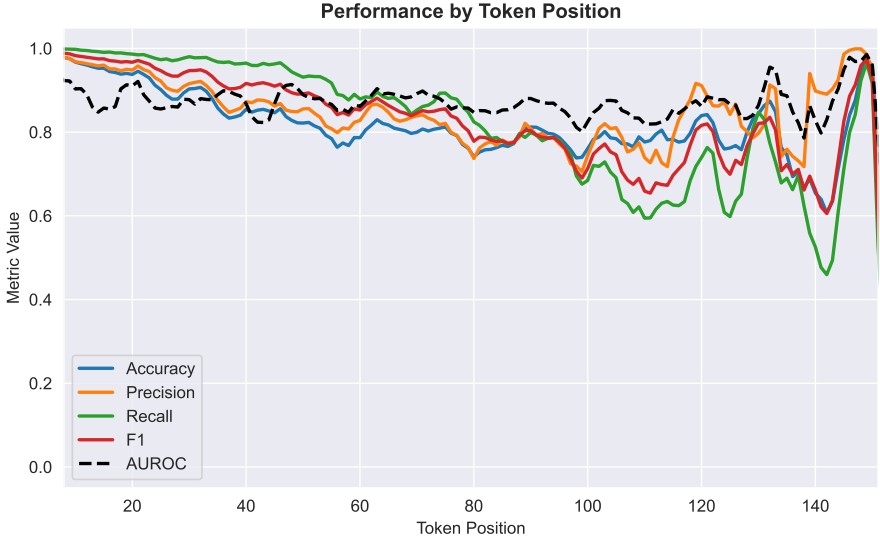

*Figure 9.* Consistent performance of HaloProbe across token positions.

We further evaluate HaloProbe on the POPE(Li et al., 2023) benchmark, which is a short-form visual question answering task. In this setting, we again treat the generated response as a binary external feature and estimate the prior using a simple tabular model. Results on COCO(Lin et al., 2014), GQA(Hudson & Manning, 2019), and AOKVQA(Schwenk et al., 2022) subsets are reported in Table 12. HaloProbe consistently improves accuracy and F1 score, showing that the proposed framework generalizes beyond object hallucination to broader discriminative settings.

## K. Reproducibility and Resources

All experiments were conducted using NVIDIA A100-SXM4-80GB GPUs. Implemented codes are available at `https://github.com/multimodal-ai-lab/HaloProbe`.

## L. Qualitative Results

In this section, we demonstrate that our method effectively removes hallucinated object tokens without affecting language fluency, even in the post-processing paradigm. This highlights the effectiveness of using HaloProbe to mitigate object

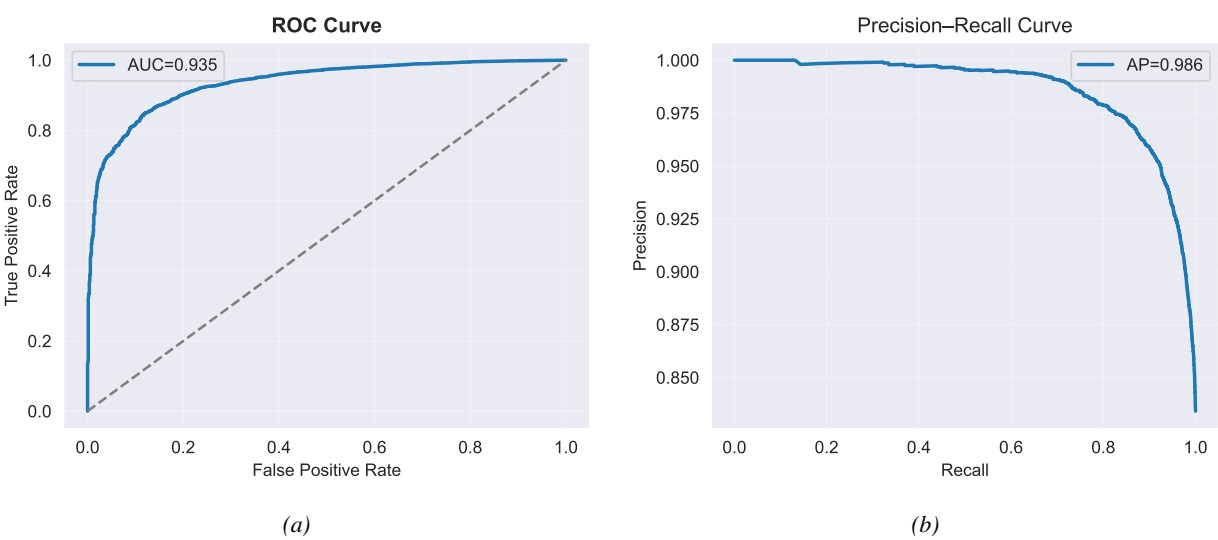

*(a)*      *(b)*

*Figure 10.* ROC and Precision-Recall curves of HaloProbe for token-level hallucination detection, illustrating performance under class imbalance.

*Table 9.* Feature ablation study for hallucination detection with HaloProbe. Internal features include attention weights and decoder logit-based signals, while external features include token position, token repetition, and token occurrence. Ablated features are replaced with Gaussian noise.

| Attntion | Logits | Token Position | Token Repettion | Token Occurrence | Accuracy ↑ | AUROC ↑ |
|:---:|:---:|:---:|:---:|:---:|:---:|:---:|
| × | ✓ | ✓ | ✓ | ✓ | 84.4 | 82.8 |
| ✓ | × | ✓ | ✓ | ✓ | 89.7 | 92.9 |
| ✓ | ✓ | × | ✓ | ✓ | 88.2 | 92.0 |
| ✓ | ✓ | ✓ | × | ✓ | 88.8 | 92.7 |
| ✓ | ✓ | ✓ | ✓ | × | 89.4 | 92.6 |
| × | × | ✓ | ✓ | ✓ | 83.9 | 81.7 |
| ✓ | ✓ | × | × | × | 88.1 | 92.1 |
| × | × | × | × | × | 84.6 | 50.1 |
| ✓ | ✓ | ✓ | ✓ | ✓ | **90.0** | **93.5** |

hallucination. Figures 11 to 13 and Figures 14 to 16 compare the Baseline with our HaloProbe + Post-Process and HaloProbe + Beam Search approaches, respectively.

*Table 10.* Effect of class balancing during training and evaluation for the internal estimator $f_\theta$. Training and testing are performed either on the natural (imbalanced) distribution or on a position-based class-balanced distribution.

| Train Balanced | Test Balanced | Accuracy ↑ | AUROC ↑ |
|:---:|:---:|:---:|:---:|
| ✗ | ✗ | 89.8 | 92.2 |
| ✓ | ✗ | 87.2 | 92.3 |
| ✗ | ✓ | 72.5 | 87.6 |
| ✓ | ✓ | 77.6 | 87.7 |

*Table 11.* Results on the AMBER benchmark using LLaVA-1.5. Higher values indicate better performance. The best results are shown in **bold**.

| Task | Method | Accuracy ↑ | Precision ↑ | Recall ↑ | F1 ↑ |
|:---|:---|:---:|:---:|:---:|:---:|
| **Relation** | Baseline | 67.4 | 83.2 | 55.2 | 66.3 |
| | **HaloProbe** | **81.2** | **88.6** | **80.2** | **84.2** |
| **Attribute** | Baseline | 75.1 | 76.0 | 81.1 | 78.5 |
| | **HaloProbe** | **88.1** | **87.6** | **88.8** | **88.2** |

*Table 12.* Results on the POPE benchmark using LLaVA-1.5. Higher values indicate better performance. The best results are shown in **bold**.

| Dataset | Method | Accuracy ↑ | Precision ↑ | Recall ↑ | F1 ↑ |
|:---|:---|:---:|:---:|:---:|:---:|
| **COCO** | Baseline | 86.8 | **94.0** | 78.7 | 85.7 |
| | **HaloProbe** | **90.7** | 93.0 | **88.1** | **90.5** |
| **GQA** | Baseline | 84.8 | 84.6 | 85.7 | 85.1 |
| | **HaloProbe** | **89.0** | **88.1** | **90.3** | **89.2** |
| **AOKVQA** | Baseline | 86.2 | 85.7 | 87.8 | 86.6 |
| | **HaloProbe** | **90.2** | **88.9** | **92.2** | **90.5** |

👤 **Prompt:** *Please describe this image in detail.*

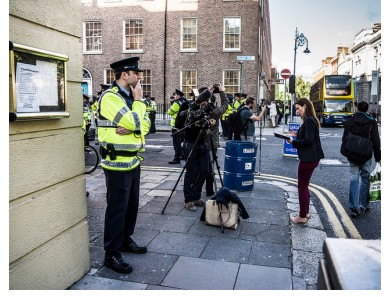

**Input Image**

⚠ **Greedy Baseline**

The image depicts a busy city street with a crowd of people walking around. Among the crowd, a woman is standing on the sidewalk, holding a cell phone and looking at it. There are several other people in the scene, some of them carrying handbags and backpacks . A police officer is standing on the side-walk, observing the crowd and ensuring public safety. A bus is visible in the background, adding to the bustling urban atmosphere. The scene also includes a traffic light , a fire hydrant , and a bicycle parked on the sidewalk.

✓ **HaloProbe + Post-Process**

The image depicts a busy city street with a crowd of people walking around. Among the crowd, a woman is standing on the sidewalk. There are several other people in the scene, some of them carrying handbags . A police officer is standing on the sidewalk, observing the crowd. A bus is visible in the background, adding to the bustling urban atmosphere.

*Figure 11.* Qualitative comparison of image description results using LLaVA-1.5-7B. The Baseline model hallucinates objects that are not present in the image, while HaloProbe + Post-Process effectively removes such hallucinations while preserving most of the correctly mentioned objects and maintaining caption fluency. Green = correct object; red = hallucinated object.

**Prompt:** *Please describe this image in detail.*

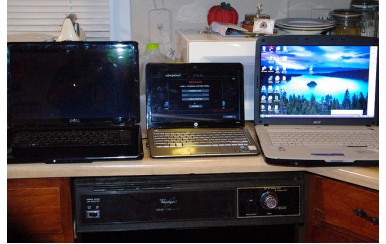

**Input Image**

⚠ **Greedy Baseline**

The image features a kitchen counter with three laptop computers placed on it. The laptops are of different sizes and are arranged in a row, with one on the left, one in the middle, and the third on the right side of the counter. In addition to the laptops , there are two keyboards on the counter, one in front of the left laptop and the other in front of the right laptop . A mouse can also be seen on the counter, positioned near the right laptop . The kitchen counter is surrounded by various items, including a bowl , a cup , and a bottle , which are placed on the countertop.

✔ **HaloProbe + Post-Process**

The image features a kitchen counter with three laptop computers placed on it.

The laptops are of different sizes and are arranged in a row, with one on the left, one in the middle, and the third on the right side of the counter.

*Figure 12.* Qualitative comparison of image description results using LLaVA-1.5-7B. The Baseline model hallucinates objects that are not present in the image, while HaloProbe + Post-Process effectively removes such hallucinations while preserving most of the correctly mentioned objects and maintaining caption fluency. Green = correct object; red = hallucinated object.

**👤 Prompt:** *Please describe this image in detail.*

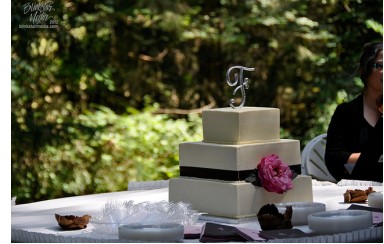

**Input Image**

**⚠ Greedy Baseline**

The image features a beautifully decorated three-tiered cake placed on a dining table . The cake is adorned with a pink rose, adding a touch of elegance to the presentation. A woman is sitting at the table, likely preparing to enjoy the cake . There are several other items on the table , including a bowl , a cup , and a spoon . A chair is positioned near the table , and a vase can be seen in the background. The scene appears to be set in a cozy and inviting environment, perfect for a celebration or gathering.

**✔ HaloProbe + Post-Process**

The image features a beautifully decorated three-tiered cake placed on a dining table . The cake is adorned with a pink rose, adding a touch of elegance to the presentation. A woman is sitting at the table , likely preparing to enjoy the cake . There are several other items on the table. It appears inviting for a celebration.

*Figure 13.* Qualitative comparison of image description results using LLaVA-1.5-7B. The Baseline model hallucinates objects that are not present in the image, while HaloProbe + Post-Process effectively removes such hallucinations while preserving most of the correctly mentioned objects and maintaining caption fluency. Green = correct object; red = hallucinated object.

**Prompt:** *Please describe this image in detail.*

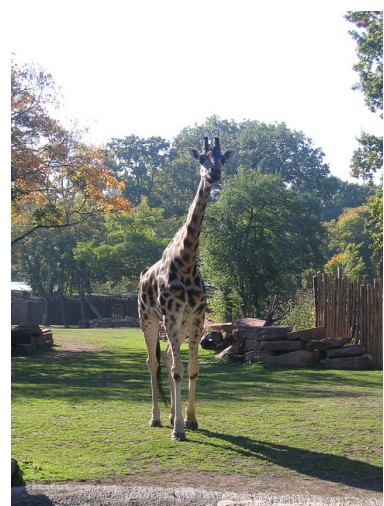

**Input Image**

⚠ **Beam Search Baseline**

The image features a giraffe standing in a grassy enclosure at a zoo. The giraffe is positioned in the center of the scene, with its long neck and legs prominently visible. The enclosure is surrounded by trees, providing a natural environment for the giraffe .

In addition to the giraffe , there are a few other elements in the scene. A truck can be seen in the background on the left side of the enclosure, possibly used for transportation or maintenance purposes. There is also a bench located near the right side of the enclosure, providing a place for visitors to sit and enjoy the view of the giraffe .

✔ **HaloProbe + Beam Search**

The image features a tall giraffe standing on a grassy field, surrounded by trees and a fence. The giraffe appears to be walking and enjoying the open area. The fence is located in the background, providing a boundary for the giraffe 's enclosure. The scene is peaceful and showcases the beauty of the giraffe in its natural habitat.

*Figure 14.* Qualitative comparison of image descriptions using LLaVA-1.5-7B. The Beam Search baseline hallucinates objects not present in the image. In contrast, HaloProbe + Beam Search selects the beam with the fewest hallucinated words and the highest number of correctly mentioned objects. Green = correct object; red = hallucinated object.

**Prompt:** *Please describe this image in detail.*

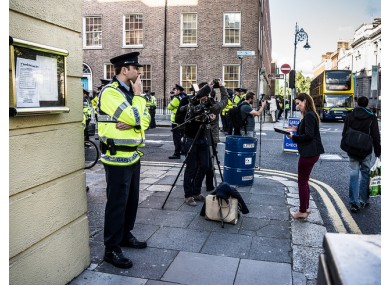

**Input Image**

⚠ **Beam Search Baseline**

The image depicts a bustling city street with a crowd of people gathered on the sidewalk. Among the crowd, there is a woman standing near the center of the scene, holding a cell phone in her hand. She appears to be engaged in a conversation or using her phone . In addition to the woman with the cell phone , there are several other people scattered throughout the scene, some closer to the foreground and others further in the background. A bus can be seen on the right side of the image, adding to the busy atmosphere of the city street.

✔ **HaloProbe + Beam Search**

The image depicts a busy city street with a crowd of people gathered around. There are several individuals standing and walking on the sidewalk, some of them carrying handbags and backpacks . A police officer is standing in the middle of the scene, likely monitoring the situation or ensuring order. There is a woman standing near the center of the scene, with a handbag placed beside her. A few other handbags can be seen among the crowd, as well as a backpack placed on the ground. In the background, a bus is visible on the right side of the scene, further emphasizing the city atmosphere. The presence of multiple people , handbags , and the police officer creates a lively and bustling urban environment.

*Figure 15.* Qualitative comparison of image descriptions using LLaVA-1.5-7B. The Beam Search baseline hallucinates objects not present in the image. In contrast, HaloProbe + Beam Search selects the beam with the fewest hallucinated words and the highest number of correctly mentioned objects Green = correct object; red = hallucinated object.

👤 **Prompt:** *Please describe this image in detail.*

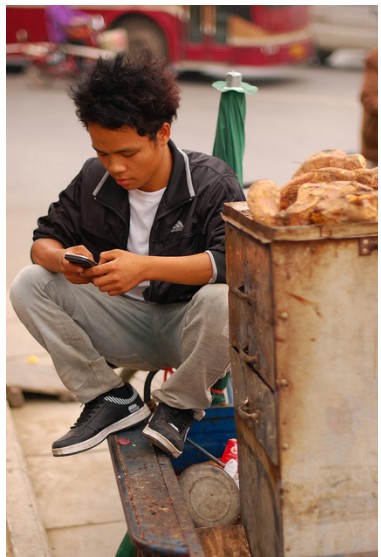

**Input Image**

⚠️ **Beam Search Baseline**

In the image, a young man is sitting on the edge of a table , looking at his cell phone . He appears to be focused on his device, possibly texting or browsing the internet. The table he is sitting on is located near the center of the scene. In the background, there are two motorcycles parked, one on the left side and the other on the right side of the image. Additionally, there are two bottles visible in the scene, one on the left side and the other on the right side of the table .

✅ **HaloProbe + Beam Search**

The image shows a man sitting on a low platform with his legs crossed. He is intently looking at his cell phone , which is placed in front of him. The man appears to be checking his phone , possibly checking messages or browsing the internet. In the background, there are two trucks visible, one on the left side and the other on the right side of the image. Another person can be seen in the background, but they are not the main focus of the scene.

*Figure 16.* Qualitative comparison of image descriptions using LLaVA-1.5-7B. The Beam Search baseline hallucinates objects not present in the image. In contrast, HaloProbe + Beam Search selects the beam with the fewest hallucinated words and the highest number of correctly mentioned objects. Green = correct object; red = hallucinated object.

