# OpenReview forum: "HaloProbe: Bayesian Detection and Mitigation of Object Hallucinations in Vision-Language Models"
_ICML.cc/2026/Conference — ICML 2026 regular_

### Official Review · Reviewer_pz67 · 2026-02-18

**Soundness:** 3
**Presentation:** 3
**Significance:** 3
**Originality:** 3
**Overall Recommendation:** 4
**Confidence:** 3

**Summary:**

This paper proposes HaloProbe, an algorithm designed to detect and mitigate hallucinations in LVLMs. The study first reveals that existing attention–based detection methods suffer from hidden confounding factors (e.g., token position and object repetition). It then introduces a factorization framework that disentangles internal decoding signals from external descriptive statistics, and combines balanced training with prior estimation to achieve reliable token-level hallucination probability estimation. The approach effectively reduces object hallucinations in image captioning.

**Compliance With Llm Reviewing Policy:**

Affirmed.

**Final Justification:**

The rebuttal has addressed my main concerns, I will maintain my positive recommendation.

**Key Questions For Authors:**

1. Please refer to the weaknesses outlined above.

2. More comprehensive related work could be discussed to better position the contributions of this paper. For example, Mixture of Decoding[1] proposes an attention-inspired adaptive decoding strategy to alleviate hallucinations, while HalluRNN[2] improves model robustness through recurrent cross-layer reasoning to mitigate hallucination phenomena, and etc.

[1] Mixture of Decoding: An Attention-Inspired Adaptive Decoding Strategy to Mitigate Hallucinations in Large Vision-Language Models.

[2] HalluRNN: Mitigating Hallucinations via Recurrent Cross-Layer Reasoning in Large Vision-Language Models

**Limitations:**

I believe that no existing method, including HaloProbe, can completely eliminate hallucinations. Therefore, the authors should explicitly acknowledge this limitation in the paper, clarifying that HaloProbe mitigates but does not fully resolve the hallucination problem, which may help set appropriate expectations for practical deployment.

**Strengths And Weaknesses:**

Strengths:

1. The motivation is clearly articulated. Figures 2-5 provide in-depth empirical analyses that lay a solid foundation for the subsequent development of HaloProbe.

2. The theoretical analysis is rigorous. By adopting a factorized Bayesian framework, the method decouples easily learnable external features from complex internal decoding signals, and employs class-conditional balanced training to mitigate shortcut learning. This design enhances interpretability and methodological soundness.

Weaknesses:

1. The LVLMs evaluated in this paper are relatively outdated and exhibit comparatively severe hallucination issues. The effectiveness of HaloProbe on more recent models, such as Qwen3-VL or Qwen2.5-VL, InternVL3.5 or InternVL3, which inherently demonstrate lower hallucination rates, remains unclear.

2. Due to the relatively limited capability of the evaluated models, the generated captions tend to be short. The generalization ability of HaloProbe to denser, more detailed captions is therefore uncertain.

3. While the method successfully reduces hallucinations, it inevitably introduces additional inference latency. It would be beneficial for the authors to include a systematic analysis of inference overhead and computational resource consumption to better inform real-world deployment considerations.

---

> ### Author Rebuttal · Authors · 2026-03-31
>
> Thank you for your helpful feedback and for recognizing the clarity of our motivation and the rigor of the Bayesian formulation, which improves interpretability and robustness to shortcut learning. We will enhance the related work with a more comprehensive discussion and clearly position our contributions relative to prior work.
>
>
>
>
>
>
>
> ## W1 - More Recent Models
>
> We acknowledge the relatively lower capability of the models used in this paper. This choice is mainly due to prior work reporting results on these models, and the difficulty of reliably reproducing those methods on newer models. However, we also evaluate newer models Qwen3-VL and InternVL3.5 with and without HaloProbe. Even when the baseline is already strong, HaloProbe still achieves a clear improvement in hallucination mitigation.
>
>
> **Qwen3-VL (default system prompt)**:
> | Decoding | Method | Chair_s $\downarrow$| Chair_i $\downarrow$ | F1 $\uparrow$|
> | --- | --- | -- | -- | -- |
> Nucleas | Baseline | 44.4 | 9.3 | 74.5 |
> Nucleas | HaloProbe + Post-process | **17.8**| **3.3** | **77** |
> Greedy | Baseline | 41.6 | 8.78 | 76.5 |
> Greedy | HaloProbe + Post-process | **16.8** | **3.5** | **78.2** |
> Beam | Baseline | 39.2 | 8.5 | 76.2 |
> Beam | HaloProbe + Beam | **23.4** | **4.9** | **76.6** |
>
>
>
>
> **Qwen3-VL (specialized system prompt for image captioning)**:
> | Decoding | Method | Chair_s $\downarrow$| Chair_i $\downarrow$| F1 $\uparrow$|
> | --- | -- | --- | -- | -- |
> Nucleas | Baseline | 25.2 | 8.4 | 74.4 |
> Nucleas | HaloProbe + Post-process | **14.4**| **4.7** | **74.5** |
> Greedy | Baseline | 24.8 | 8 | **74.6** |
> Greedy | HaloProbe + Post-process | **12.2** | **4.3** | **74.6**|
> Beam | Baseline | 25.8 | 7.9 | 74.9 |
> Beam | HaloProbe + Beam | **15.6** | **4.9** | **75.2** |
>
>
>
>
> **InternVL3.5**:
> | Decoding | Method | Chair_s $\downarrow$| Chair_i $\downarrow$| F1 $\uparrow$|
> | ---- | -- | --- | --- | -- |
> Nucleas | Baseline | 33.8 | 9.6 | 73.8 |
> Nucleas | HaloProbe + Post-process | **22.2**| **6.7** | **75.1** |
> Greedy | Baseline | 31.6 | 8.7 | 74.5 |
> Greedy | HaloProbe + Post-process | **16.6** | **5.1** | **75.7** |
> Beam | Baseline | 34.6 | 9.5 | 73.7 |
> Beam | HaloProbe + Beam | **15** | **5.2** | **74** |
>
>
>
>
>
>
>
> &nbsp;
> ___
>
>
> ## W2 - Robustness to Caption Length
>
>
> In W1, we provide results on Qwen3-VL with both its default system prompt and a modified version. The average caption length in the default setting is ~300 tokens, which is about 3× longer than in LLaVA-1.5 and also Qwen3-VL with a specialized captioning prompt. Despite this difference in token length, HaloProbe preserves its effectiveness and improves over the baseline with a notable margin.
>
>
>
> &nbsp;
> ___
>
> ## W3 - Inference Latency
>
>
> Thank you for raising this point, and we will include the inference latency analysis in the paper.
>
> HaloProbe is a shallow MLP classifier that adds marginal latency to the original decoding process of the LVLM. On an NVIDIA A100-SXM4-80GB with LLaVA-1.5 7B, the average latency of the data preparation step is 32 ms and the MLP computation is 17 ms, totaling ~49 ms. In contrast, LVLM inference takes around 2446 ms (\~50x). Therefore, in downstream mitigation tasks, the latency of HaloProbe is negligible, and the remaining cost comes from other components:
> - In the beam search strategy, latency is close to the standard beam search baseline if implemented efficiently.
> - In the post-processing strategy, the external LLM or other NLP-based editing modules add a constant overhead to the LVLM inference latency.

---

> > ### Author Rebuttal · Reviewer_pz67 · 2026-04-03
> >
> > Thanks for the author's response. I will maintain my positive recommendation.

---

> > > ### Author Response · Authors · 2026-04-06
> > >
> > > We sincerely thank the reviewer for the positive feedback and for acknowledging that their concerns have been fully resolved. We kindly invite the reviewer to consider reflecting this updated assessment in their final score, if appropriate. We would greatly appreciate your support.

---

### Official Review · Reviewer_oKAV · 2026-03-12

**Soundness:** 3
**Presentation:** 3
**Significance:** 3
**Originality:** 3
**Overall Recommendation:** 4
**Confidence:** 3

**Summary:**

This work introduces HaloProbe, a Bayesian framework designed to estimate token-level hallucination probabilities by decomposing external descriptive statistics from internal decoding signals. By employing balanced training to isolate internal evidence and integrating it with learned external feature priors, HaloProbe effectively recovers the true posterior probability of hallucinations.

**Compliance With Llm Reviewing Policy:**

Affirmed.

**Final Justification:**

The rebuttal has addressed my main concerns. I lean toward a Weak Accept.

**Key Questions For Authors:**

* Will balanced training lead to overfitting due to the oversampling of minority hallucinations?
* Can the influence of more high-dimensional confounding factors be incorporated into the framework of this paper? Such as the visual complexity of images, or the distribution bias of the model's training data.

**Limitations:**

Yes

**Strengths And Weaknesses:**

### Strength
* The paper keenly identifies a potential bias in prior research, specifically challenging the claim that "hallucinated objects consistently exhibit lower image attention." The authors demonstrate that token position and object repetition are often hidden confounding factors, providing a more nuanced understanding of the phenomenon.

* The framework elegantly decouples internal decoding signals (e.g., fine-grained attention, logit features) from external statistical priors (e.g., location, frequency). Through balanced training and posterior calibration, it effectively mitigates the severe imbalance of object hallucination samples, preventing the model from falling into the trap of shortcut learning.

### Weakness
* The manuscript currently lacks an analysis of inference-time overhead. Since HaloProbe-guided beam search incurs a computational cost that scales linearly with beam size, a comprehensive latency study is essential to assess its practical utility in real-time applications.
* By framing hallucination detection as a token-based binary classification task, the model struggles with non-atomic errors. It lacks the granularity required to address attribute binding or relational hallucinations (e.g., "a red cat" where the object is present but the attribute is hallucinated), as the current analysis is restricted to individual object tokens.

---

> ### Author Rebuttal · Authors · 2026-03-31
>
> We thank the reviewer for recognizing our contributions, including the analysis of bias in prior attention-based methods, the identification of token position and repetition as confounders, and our factorized framework that decouples internal signals from external priors through balanced training.
>
>
>
> ## W1 - Inference Time
>
> We thank the reviewer for raising this point.
>
> First, we clarify that the implementation novelty of the mitigation part (beam search and post-processing) is not among the main contributions of this work. In this part, our main goal is to show that, given an accurate hallucination detector, even simple mitigation strategies can achieve state-of-the-art performance. We also clarify that Table 4 compares the beam search variant of HaloProbe with the beam search variants of other methods under comparable settings.
>
> Regarding runtime, the detection module is lightweight. The classifier is a shallow MLP, and its FLOPs are negligible compared to standard LVLM decoding. On an NVIDIA A100-SXM4-80GB with LLaVA-1.5 7B, the average inference time is:
>
> - **LVLM decoding: 2446 ms**
> - **Feature extraction part of HaloProbe: 32 ms**
> - **Classification part of HaloProbe: 17 ms**
>
> Thus, the overhead from detection can be ignored in practice. Concretely for the mitigation part:
> - Post-processing requires one LVLM forward pass (equivalent to the simple baseline) and one forward pass of the LLM editor.
> - HaloProbe-guided beam search has complexity similar to standard beam search in the baseline when implemented efficiently, with negligible overhead from the HaloProbe detector.
>
> Last but not least, HaloProbe enables more efficient search strategies. For example, a branch-and-bound approach can be implemented:
> - terminate early when a hallucination-free branch is found and move to the next stage
> - stop generating low-quality branches (e.g., when more than one hallucinated object is observed in the branch)
>
>
> This can reduce complexity below linear in beam size. While promising, the efficiency optimization is not the focus of this study and is left for future work.
>
>
>
> &nbsp;
> ___
>
>
> ## W2 - Attribute and Relation Hallucination
> The main focus of this study is object hallucination, which is the common setting in prior work. While other types of hallucination are beyond the initial scope of this paper, the HaloProbe framework can be adapted to these cases, including relation and attribute hallucination.
>
> To support this, we evaluate HaloProbe on the **relation** and **attribute** hallucination subsets of the AMBER benchmark using LLaVA-1.5. Since this is a discriminative setting, we redefine the external features and use the model’s generated response as a binary external feature. In this setting, mitigation is simply performed by selecting the final decision based on the detector prediction. The results show a notable improvement of HaloProbe in these settings.
>
>
> **Relation**:
> |Method|Accuracy|Precision|Recall|F1
> |-|-|-|-|-|
> Baseline|67.4|83.2|55.2|66.3
> HaloProbe|81.2|88.6|80.2|84.2
>
> **Attribute**:
> |Method|Accuracy|Precision|Recall|F1
> |-|-|-|-|-|
> Baseline|75.1|76.0|81.1|78.5
> HaloProbe|88.1|87.6|88.8|88.2
>
>
> &nbsp;
> ___
>
>
> ## Q1
>
> We train a shallow MLP as the main estimator using \~2.5k training captions (each containing multiple hallucinated and correct objects). Moreover, our token-wise balancing uses a window over positions and computes class ratios within each window (similar to Figure 5, but with overlapping windows). This avoids extreme oversampling when minority samples are scarce at specific positions. We also clip the oversampling ratio at 15 and use a low number of training epochs (~10).
> Overall, the risk of overfitting is low due to the limited model capacity, sufficient data, window-based smoothing, ratio clipping, and few training epochs.
>
>
> &nbsp;
> ___
>
>
> ## Q2
>
> Defining external features depends on the problem setting. Low-dimensional features are preferred, since we can balance the training set of \( f_\theta \) conditioned on their values. For example, if a model behaves differently on outdoor vs. indoor images, this can be conditioned as a binary confounder, and the dataset can be balanced by oversampling the minority group.
>
> For high-dimensional factors (e.g., visual complexity or training data bias), conditioning and approximate prior learning are still possible. However, conditional dataset balancing becomes less straightforward in such cases.

---

> > ### Author Rebuttal · Reviewer_oKAV · 2026-04-03
> >
> > Thanks for the author's response.  I will maintain my positive score.

---

> > > ### Author Response · Authors · 2026-04-06
> > >
> > > We sincerely thank the reviewer for the positive feedback and for acknowledging that their concerns have been fully resolved. We kindly invite the reviewer to consider reflecting this updated assessment in their final score, if appropriate. We would greatly appreciate your support.

---

### Official Review · Reviewer_DNBA · 2026-03-13

**Soundness:** 2
**Presentation:** 2
**Significance:** 2
**Originality:** 3
**Overall Recommendation:** 4
**Confidence:** 3

**Summary:**

The paper addresses object hallucination detection and mitigation in Large Vision-Language Models (LVLMs). The authors propose HaloProbe, a Bayesian detection framework that factorizes internal model signals (e.g., attention, logits) from external caption statistics (e.g., position, repetition) using a balanced training strategy and a learned prior. Applying HaloProbe for non-invasive mitigation using hallucination-aware beam search and post-hoc editing, achieving state-of-the-art results on standard benchmarks across multiple LVLMs without altering the models' internal dynamics.

**Compliance With Llm Reviewing Policy:**

Affirmed.

**Final Justification:**

My concerns are fully resolved and I have raised the scores accordingly.

**Key Questions For Authors:**

1. How's HaloProbe-guided beam search compare to the lightweight, intervention-based methods (like PAI) during inference?
2. The framework explicitly models object repetition and token position as external features. How well does HaloProbe generalize to out-of-distribution text lengths or entirely different prompting structures (e.g., short-form VQA vs. long-form detailed descriptions)?

**Limitations:**

yes

**Strengths And Weaknesses:**

Strengths
1. The analysis is insightful. The identification of Simpson's paradox in attention-based hallucination detection is a strong, novel contribution.
2. The proposed Bayesian formulation cleanly factorizes the problem, effectively handling the severe class imbalance between hallucinated and correct tokens.
3. HaloProbe achieves state-of-the-art performance on hallucination detection and mitigation benchmarks across diverse models like LLaVA-1.5, Shikra, and MiniGPT-4

Weakness
1. The models being studied are too old. Demonstrating the author's approach on recent models such as Qwen3-VL would strength the method's effectiveness.
2. The method is limited to object hallucination problems only, while there are more challenging scenarios including attribute and relational hallucinations.

---

> ### Author Rebuttal · Authors · 2026-03-31
>
> Thank you for your valuable feedback and for highlighting the novelty of our Simpson’s paradox analysis, the effectiveness of the Bayesian factorization, and the strong state-of-the-art results.
>
>
> ## W1 - More Recent Models
> We agree with your concern about evaluating newer models. This was also a priority for us; however, a key limitation is the lack of reported results from competing baselines on these newer models, which makes direct comparison difficult. Re-implementing all baselines on new backbones is also challenging, so we used commonly studied models from prior work.
>
>
> That said, we can still evaluate each model when equipped with HaloProbe. We report the performance of recent models, **Qwen-3VL** and **InternVL-3.5**, with and without HaloProbe, under different decoding regimes (similar to Table 4). We will include these comparisons with more recent models in the revised version. HaloProbe consistently reduces hallucinations on both models (e.g., Chair_s drops from 44.4 → 17.8 on Qwen3-VL), confirming effectiveness on newer LVLMs. [Full tables are reported under W1 response of reviewer **pz67**.]
>
>
>
> &nbsp;
> ___
>
>
> ## W2 - Attribute and Relation Hallucination
>
> In this work, we focus on object hallucination (as stated in the name of the paper) since our primary setting is open-ended caption generation, which aligns with our confounding analysis. MS COCO provides the most suitable benchmark with object-level annotations. Moreover, most prior work we compare against also focuses on object hallucination, making this a common setting.
> However, our proposed framework is easily applicable to other scenarios. The following tables include evaluation on the attribute and relation benchmarks from the AMBER dataset, performed on LLaVA-1.5:
>
>
> **Relation**:
> |Method|Accuracy|Precision|Recall|F1
> |-|-|-|-|-|
> Baseline|67.4|83.2|55.2|66.3
> HaloProbe|81.2|88.6|80.2|84.2
>
> **Attribute**:
> |Method|Accuracy|Precision|Recall|F1
> |-|-|-|-|-|
> Baseline|75.1|76.0|81.1|78.5
> HaloProbe|88.1|87.6|88.8|88.2
>
>
>
>
> &nbsp;
> ___
>
>
> ## Q1
>
>
> The HaloProbe detector is a shallow MLP applied to noun tokens in the caption (or sub-caption in beam search) in a batch-wise manner. Therefore, its computational overhead is negligible compared to the LVLM.
> For each caption, the average computation time of HaloProbe is about 49 ms, while the average latency of the LVLM (LLaVA-1.5) is about 2446 ms on a NVIDIA A100-SXM4-80GB. This efficiency also extends to the beam search setting when implemented with proper KV caching, shared computation, and batched beam processing. In an efficient implementation, the overall cost is close to that of standard beam search.
>
> We also clarify that Table 4 compares beam search variants of all baselines under similar settings.
>
>
> &nbsp;
> ___
>
>
> ## Q2
>
> The HaloProbe framework is general and can be applied across different settings, including long-form detailed captions and short-answer benchmarks. Table 1 of W1 response of reviewer **pz67** includes the performance of Qwen3-VL under the default system prompt with an average length of ~300 tokens, which is ~3× longer than the captions analyzed in this paper.
>
> To further support this, we include results on the POPE benchmark, which is a short-form discriminative VQA task. In this setting, we redefine the external features as a binary generated response and learn the prior using a tabular approach instead of a neural network.
>
>
> **COCO**:
> |Method|Accuracy|Precision|Recall|F1|
> |-|-|-|-|-|
> Baseline|86.8|94.0|78.7|85.7|
> HaloProbe|90.7|93.0|88.1|90.5|
>
>
> **GQA**:
> |Method|Accuracy|Precision|Recall|F1|
> |-|-|-|-|-|
> Baseline|84.8|84.6|85.7|85.1|
> HaloProbe|89.0|88.1|90.3|89.2|
>
>
> **AOKVQA**:
> Method|Accuracy|Precision|Recall|F1|
> |-|-|-|-|-|
> Baseline|86.2|85.7|87.8|86.6|
> HaloProbe|90.2|88.9|92.2|90.5|
>
>
> **Out-of-distribution generalization:** In the case where the evaluation distribution differs from the training distribution, preserving performance is not guaranteed. However, the factorized form of HaloProbe provides better robustness in such cases. If an external feature (e.g., token position) becomes a spurious signal under distribution shift, its effect can be reduced in the prior while still conditioning $f_\theta$ on it. In this way, we retain its predictive value while controlling its spurious influence in both representation learning and inference.

---

> > ### Author Rebuttal · Reviewer_DNBA · 2026-03-31
> >
> > My concerns are fully resolved and I have raised the scores accordingly.

---

> > > ### Author Response · Authors · 2026-04-07
> > >
> > > We sincerely thank the reviewer for the positive feedback and for acknowledging that the concerns have been fully resolved. We are pleased that our clarifications have fully addressed the issues.

---

### Official Review · Reviewer_RDbh · 2026-03-15

**Soundness:** 3
**Presentation:** 3
**Significance:** 2
**Originality:** 2
**Overall Recommendation:** 4
**Confidence:** 3

**Summary:**

This paper targets object hallucinations in LVLMs and proposes HaloProbe, a Bayesian detection and non-invasive mitigation framework. The authors show that relying on attention strength alone suffers from hidden confounders (token position and object repetition), leading to Simpson’s paradox. HaloProbe disentangles internal signals (attention/logit confidence) and external statistics (position/repetition) via a balanced estimator and a prior estimator, recovering posterior hallucination probabilities. The score is then used for mitigation with hallucination-aware beam search or post-processing via an external LLM. Experiments are on MS COCO with CHAIR, using 7B LLaVA-1.5, Shikra, and MiniGPT-4.

**Compliance With Llm Reviewing Policy:**

Affirmed.

**Final Justification:**

The authors' rebuttal fully addresses my concerns. I will therefore raise my score.

**Key Questions For Authors:**

1. What are the detection and mitigation inference overheads?

2. The post-processing editor introduces an external LLM. Can you achieve similar gains without it (e.g., rule-based or self-model editing)?
Have you tested on broader tasks (VQA, multi-turn dialogue, attribute/relational hallucinations)?

3. The models used in the experiments are outdated. Multimodal models like LLaVA and MiniGPT-4 are older-generation LVLMs. The latest multimodal models (e.g., Qwen3.5, GLM-4.7) might not suffer from these issues as severely.  Would the latest multimodal models suffer from such object hallucinations? Would the method gain performance on these models?

**Limitations:**

Yes.

**Strengths And Weaknesses:**

Strengths:
1. This paper Identifies and analyzes Simpson’s paradox caused by position/repetition confounders, explaining why coarse attention statistics can be misleading.
2. The Bayesian factorization cleanly separates internal evidence from external priors, conceptually improving robustness to bias and imbalance.
3. Strong detection gains over several baselines (IC, UT, EAZY, Devils), and consistent mitigation improvements across models and decoding strategies.

Weaknesses:
1. The main novelty lies in the confounding analysis and Bayesian decomposition; the mitigation components (beam re-ranking and post-hoc editing) are close to existing decoding/post-hoc strategies. The paper should clarify the core differentiators against prior work.
2. The post-processing mitigation depends on an external LLM (GPT‑5 in the example), raising reproducibility, cost, and fairness concerns. There is no ablation or fair comparison without external editors.
3. Evaluation is limited to MS COCO and object hallucinations. Broader datasets (e.g., Flickr30k, NoCaps) or tasks with attribute/relational hallucinations are missing.
4. Inference cost and latency are under-discussed. The detector uses top‑K attention statistics and logit features; mitigation further adds beam search or second-pass editing.

---

> ### Author Rebuttal · Authors · 2026-03-31
>
> Thank you for your thoughtful feedback and for highlighting our analysis of Simpson’s paradox, the Bayesian factorization framework, and the strong empirical improvements.
>
> ## W1
> We agree that the main *methodological* contribution of this paper lies in the detection component (HaloProbe). While we emphasize this in the manuscript, we will provide a more detailed elaboration. Beyond the methodological aspects, both detection and mitigation provide important insights and motivation:
> - In the detection part, we analyze hidden confounders and challenge the common assumption that hallucinated objects receive less image attention.
> - In the mitigation part, we highlight a limitation of recent intervention-based methods, which can lead to unintended side effects.
>
> We believe these insights help improve the reliability of methods in the field, beyond introducing new techniques.
>
> Additionally, the mitigation component serves to demonstrate the effectiveness of HaloProbe in a downstream setting and enables comparison with prior work. It shows that even simple, non-invasive strategies can achieve SOTA performance when guided by an accurate hallucination detector.
>
>
> ## W2 & Q2
> Thank you for raising this point. In addition to the clarification in W1 regarding the mitigation section, we clarify that the external LLM editor is only used to preserve linguistic fluency in the post-processing method. Since it is not equipped with the image, it cannot improve the mitigation performance of HaloProbe and impact the fairness of the comparison in Table 4.
>
> Moreover, the LLM editor is not a required component of our post-processing pipeline. Following your suggestion, we implemented a rule-based editing pipeline with up to 15 simple grammatical rules to remove marked objects. This process is fully automatic and uses standard NLP tools such as POS tagging. The table below compares the performance of the rule-based editor and the external LLM editor on LLaVA-1.5.
>
>
> |Method|Chair_s|Chair_i|F1
> |-|-|-|-|
> External LLM (GPT5)|17.6|5.2|75.2
> **Rule-Based**|17|5.4|75.3
>
> The rule-based editor matches the external LLM, showing our method remains competitive without relying on an external LLM.
>
> The rules are derived automatically from the structure of marked sentence examples in a holdout subset of captions. For example:
>
> - In “and” pairs, remove only the hallucinated object, keeping the rest (e.g., “a dog and a $cat” → “a dog”).
>
> - In existential sentences, remove the whole sentence if the subject is hallucinated (e.g., “there is a $bench…”).
>
>
>
> ## W3 & Q2
>
> We acknowledge the importance of broader evaluation. However, hallucination evaluation itself is challenging and still an active research area. One key issue is handling synonym groups for object classes in open-ended captioning. For example, if the model outputs *person* instead of the annotated *man*, it may be incorrectly counted as hallucination. Many datasets (e.g., Flickr30k) are not well-suited for this, while MS COCO mitigates this issue by defining synonym classes, despite being limited to 80 categories.
>
> To demonstrate broader applicability, we evaluate on the **AMBER dataset**, which includes **relation** and **attribute** hallucination. We use LLaVA-1.5 and incorporate the same internal features (attention and logit scores) for HaloProbe, while defining a simple external feature based on the model’s binary prediction (yes/no). Results are reported under the W2 response of reviewer **DNBA**.
>
> These results show the effectiveness of HaloProbe for relation and attribute hallucination, and that it can be easily adapted to discriminative settings. While this is beyond the initial scope of the paper, which focuses on object hallucination, we will include this evaluation in the revised version.
>
> ## W4 & Q1
>
> Thank you for highlighting this point. HaloProbe consists of a lightweight MLP, and its overhead is minor compared to the base LVLM. For captions generated by LLaVA-1.5, the average inference time is **2446 ms** per sample, while HaloProbe adds only **49 ms** of latency.
>
> Beam search introduces latency similar to standard beam search baselines when implemented efficiently. In post-processing with an external LLM, the second-pass editing adds another generation step, making the total cost roughly twice that of a single LVLM pass. However, this editor is not mandatory and can be replaced with a lightweight refinement module, such as the rule-based strategy discussed in W2, when latency is a concern.
>
> ## Q3
> We acknowledge the lack of newer multimodal models. This is mainly due to the absence of reported results from prior methods on these models. However, we can evaluate newer models and compare their hallucination rates with and without HaloProbe. The hallucination evaluation on Qwen3-VL and InternVL3.5 models are reported under the W1 response of reviewer **pz67**.
>
> Due to the time and space limits of the rebuttal, we will consider additional models in the camera-ready version.

---

> > ### Author Rebuttal · Reviewer_RDbh · 2026-04-04
> >
> > Thanks for your response.
> >
> > Regarding W1, the explanation provided is not sufficiently thoughtful.
> >
> > Regarding W4 & Q1, is the latency comparison based on beam search? Beam search introduces multiple branches and is slower, and is often not applied in real-time applications. Can you provide decoding time comparison results under greedy decoding (or sampling)?

---

> > > ### Author Response · Authors · 2026-04-06
> > >
> > > Thank you for your continued engagement and for encouraging us to clarify previously unclear points.
> > >
> > > Regarding W1, we think the character limit of the rebuttal restricted the clarity of our previous response, and we now provide a more detailed explanation.
> > >
> > > We again restate that we agree with your comment about the methodology of the mitigation part and clarify that we did not claim methodological novelty for it. In multiple parts of the paper (listed below), we focus on two aspects instead: 1) being non-invasive and close to the model's standard decoding, and 2) reflecting the SOTA performance of the HaloProbe detector. Based on your suggestion, **we will explicitly mention in the revised version** that the main role of the mitigation part is these aspects, **not offering a novel mitigation method**.
> > >
> > > As you and other reviewers noted, the main contribution lies in the Bayesian detector and statistical analysis (e.g., Simpson's paradox).
> > >
> > > Relevant parts of the paper:
> > >
> > > 1. Abstract, lines 32, 33, 36: emphasize non-invasive and SOTA performance of mitigation rather than methodology.
> > > 2. Introduction, line 103: “We show that HaloProbe can be employed as an effective probe for non-invasive post-hoc mitigation methods.”
> > > 3. Contributions, lines 71-75: mitigation emphasizes HaloProbe’s role, not algorithmic novelty.
> > > 4. Hallucination Mitigation, lines 275-277: “Importantly, the decoding process itself is unchanged; HaloProbe is used only as an external scoring mechanism.”
> > > 5. Experiments, lines 409-412: “Standard LVLM decoding, when guided by an external scoring signal such as HaloProbe, is sufficient to generate fluent and accurate captions.”
> > >
> > >
> > >
> > >
> > > &nbsp;
> > > ___
> > >
> > >
> > > Regarding W4, we clarify that our reported numbers for inference time were measured under standard greedy decoding (same as sampling). A vanilla greedy pass of LLaVA-1.5 on an NVIDIA A100-SXM4-80GB takes **2446 ms**, while generating a caption with HaloProbe takes **2495 ms**, adding only **49 ms** for hallucination detection.
> > >
> > > We agree that beam search introduces additional overhead. Inference time under beam search depends strongly on implementation optimizations rather than FLOPs alone. Using an efficient beam-search implementation, the average latency with beam size 5 is **4652 ms** for the baseline and **5027 ms** with HaloProbe. For beam size 2, the results are **3791 ms** and **4126 ms**, respectively. Notably, these results show that beam search does not necessarily scale linearly with the number of beams when using proper KV-caching and batching.
> > >
> > > Finally, while computational efficiency was not the primary goal of this paper, these results demonstrate the low overhead and practical efficiency of HaloProbe.

---

### Decision · Program_Chairs · 2026-04-30

**Decision:**

Accept (regular)

**Comment:**

This paper studies object hallucination in vision-language models and proposes HaloProbe, a Bayesian framework for detection and mitigation that combines internal decoding signals with external prior statistics. The reviewers generally found the paper technically sound, well motivated, and supported by solid empirical evidence. A particular strength repeatedly noted in the discussion is the paper’s analysis of confounding effects in attention-based detection, especially the Simpson’s paradox perspective, which helps explain why prior token-level signals can be misleading. The main concerns were about the exact novelty relative to prior detection methods, the dependence of the mitigation pipeline on external editing modules, the limited coverage of newer models and broader hallucination settings, and the practical overhead of the proposed approach.

In my view, the authors handled these concerns well in the rebuttal. They clarified that the main contribution is the detector and its Bayesian factorization rather than the editing module alone, added stronger discussion and comparisons for mitigation without external LLM editing, expanded results to newer models and attribute/relation hallucination, and provided additional latency analysis. These clarifications substantially improved the paper. Some limitations remain, so this is not a strong accept, but the contribution is meaningful and the work is careful and solid. I therefore recommend Weak Accept.